# Coastal-ocean uptake of anthropogenic carbon

Timothée Bourgeois[1], James C. Orr[1], Laure Resplandy[2], Jens Terhaar[1], Christian Ethé[3], Marion Gehlen[1], and Laurent Bopp[1]

[1]Laboratoire des Sciences du Climat et de l'Environnement, LSCE/IPSL, CEA-CNRS-UVSQ, Université Paris-Saclay, F-91191 Gif-sur-Yvette, France
[2]Scripps Institution of Oceanography, University of California San Diego, La Jolla, CA, USA
[3]Institut Pierre Simon Laplace, 4 Place Jussieu, 75005 Paris, France

*Correspondence to:* T. Bourgeois (timothee.bourgeois@lsce.ipsl.fr)

**Abstract.** Anthropogenic changes in atmosphere-ocean and atmosphere-land $CO_2$ fluxes have been quantified extensively, but few studies have addressed the connection between land and ocean. In this transition zone, the coastal ocean, spatial and temporal data coverage is inadequate to assess its global budget. Thus we use a global ocean biogeochemical model to assess the coastal ocean's global inventory of anthropogenic $CO_2$ and its spatial variability. We used an intermediate resolution, eddying version of the NEMO-PISCES model (ORCA05), varying from 20 to 50 km horizontally, i.e., coarse enough to allow multiple century-scale simulations but finer than coarse resolution models ($\sim$200 km) to better resolve coastal bathymetry and complex coastal currents. Here we define the coastal zone as the continental shelf area, excluding the proximal zone. Evaluation of the simulated air-sea fluxes of total $CO_2$ for 45 coastal regions gave a correlation coefficient R of 0.8 when compared to observation-based estimates. Simulated global uptake of anthropogenic carbon results averaged $2.3 \, \mathrm{Pg\,C\,yr^{-1}}$ during 1993–2012, consistent with previous estimates. Yet only $0.1 \, \mathrm{Pg\,C\,yr^{-1}}$ of that is absorbed by the global coastal ocean. That represents 4.5% of the anthropogenic carbon uptake of the global ocean, less than the 7.5% proportion of coastal-to-global ocean surface areas. Coastal uptake is weakened due to a bottleneck in offshore transport, which is inadequate to reduce the mean anthropogenic carbon concentration of coastal waters to the mean level found in the open-ocean mixed layer.

## 1 Introduction

The ocean mitigates climate change by absorbing atmospheric $CO_2$ produced by combustion of fossil fuels, land-use change, and cement production. During 2005–2014, the global ocean absorbed $2.6 \pm 0.5 \, \mathrm{Pg\,C\,yr^{-1}}$ of anthropogenic carbon, an estimated 26% of the total anthropogenic $CO_2$ emissions (Le Quéré et al., 2015). The global anthropogenic carbon budget relies on separate estimates for atmosphere, land, and ocean reservoirs. Yet it neglects what happens in the aquatic continuum between land and ocean (Cai, 2011; Regnier et al., 2013), for which there is no consensus on anthropogenic carbon uptake (Wanninkhof et al., 2013; Mackenzie et al., 2004; Bauer et al., 2013; Regnier et al., 2013; Le Quéré et al., 2015; Ciais et al., 2013).

The land-ocean aquatic continuum includes inland waters, estuaries, and the coastal ocean, i.e., the succession of active physical- biogeochemical systems that connect upland terrestrial soils to the open ocean (Regnier et al., 2013). Our focus here is on the coastal ocean, which plays an inordinately large role relative to the open ocean in terms of primary productivity, export

production and carbon burial. Although the coastal ocean covers only 7-10% of the global ocean surface area, it accounts for up to 30% of oceanic primary production, 30-50% of oceanic inorganic carbon burial, and 80% of oceanic organic carbon burial (Gattuso et al., 1998; Longhurst et al., 1995; Walsh, 1991); moreover, the coastal ocean supplies about half of the organic carbon that is delivered to the deep open ocean (Liu et al., 2010). All these estimates suffer from high uncertainties as do those for coastal-ocean air-sea $CO_2$ exchange (Laruelle et al., 2014), particularly its anthropogenic component. Indeed, in addition to the effect of increasing atmospheric $CO_2$, potential changes in coastal ocean physics (e. g., temperature) and biology (e.g., net ecosystem production) as well as changes in riverine input and interactions with the sediment may be of primary importance (Mackenzie et al., 2004; Hu and Cai, 2011). These changes would modify the distribution of carbon and alkalinity, and hence change the potential of the coastal ocean to absorb anthropogenic carbon.

To date, few studies have distinguished anthropogenic carbon uptake by the global coastal ocean. Estimating air-sea fluxes of anthropogenic $CO_2$ in the coastal ocean would require multi-decadal time-series of coastal $CO_2$ observations in order to extract an anthropogenic signal from the strong coastal natural variability. Such time-series are still rare and probably not long enough. To our knowledge, the only available equivalent time-series are the Ishii et al. (2011) 1994-2008 time series along 137°E on Japanese coasts and the Astor et al. (2013) 1996-2008 time-series at the CARIACO station on Venezuelan coasts. Therefore, estimates of anthropogenic carbon uptake by the global coastal ocean rely mainly on modelling, extrapolations from the open-ocean and/or closing- or/ balanced- budget approaches. An early modelling approach was proposed by Andersson and Mackenzie (2004) and Mackenzie et al. (2004). They used a 2-box model (Shallow-water Ocean Carbonate Model, SOCM) that separated the coastal ocean into surface waters and sediment pore waters. They estimated that the preindustrial coastal ocean was a source of $CO_2$ to the atmosphere and had recently switched to a $CO_2$ sink. This source-to-sink switch is mainly caused by a shift in net ecosystem production (NEP) due to increased anthropogenic nutrient inputs (Andersson and Mackenzie, 2004; Mackenzie et al., 2004). Another proposed mechanism is simply linked to the anthropogenic increase in atmospheric $CO_2$, considering constant NEP (Bauer et al., 2013). The difference between the simulated air-sea $CO_2$ fluxes from the SOCM model for years 1700 and 2000 suggests that in 2000 the coastal ocean absorbed $0.17\,\mathrm{Pg\,C\,yr^{-1}}$ of anthropogenic carbon from the atmosphere (Borges, 2005). As for extrapolation, Wanninkhof et al. (2013) used coarse-resolution global-ocean models and observations and estimated a similar uptake of $0.18\,\mathrm{Pg\,C\,yr^{-1}}$ by extrapolating open-ocean air-sea fluxes of anthropogenic $CO_2$ into the coastal zone . Finally, Liu et al. (2010) combined estimates from the same SOCM model for the preindustrial coastal zone with observational estimates of the contemporary flux to deduce a corresponding anthropogenic carbon uptake of $0.5\,\mathrm{Pg\,C\,yr^{-1}}$ for the 1990s.

In addition, there exist 3-D regional circulation-biogeochemistry-ecosystem models that have been used to study other aspects of coastal ocean carbon cycling as summarized by Hofmann et al. (2011). Typically, such models have been implemented in regions where sufficient measurements are available for model validation, e.g., the Middle Atlantic Bight (eastern U.S. coast) (Fennel et al., 2008; Fennel, 2010), the California Current System (Fiechter et al., 2014; Turi et al., 2014; Lachkar and Gruber, 2013), and the European shelf seas (Artioli et al., 2014; Phelps et al., 2014; Wakelin et al., 2012; Allen et al., 2001; Cossarini et al., 2015; Prowe et al., 2009). Because of their limited regional domains, such models are typically able to make simulations with horizontal resolutions of 10 km or less, which remains a challenge for global circulation-biogeochemical models. The

reduced computational requirements of regional models also allows biogeochemistry and ecosystem components to be more complex. Unfortunately, joining together a network of regional models to allow efficient simulations that cover all parts of the global coastal ocean remains a technical challenge (Holt et al., 2009).

The alternative of using a global model is computationally more challenging because few of them have adequate resolution to properly simulate many critical coastal-ocean processes (Griffies et al., 2010; Holt et al., 2009). Coarse-resolution global models fail to adequately resolve the coastal bathymetry, which substantially alters coastal ocean circulation (Fiechter et al., 2014) as well as mesoscale dynamics, upwelling, and coastal currents, all of which are thought to strongly affect the variability of air-sea $CO_2$ fluxes along ocean margins (Borges, 2005; Lachkar et al., 2007; Kelley et al., 1971). Global models also typically lack a benthic component, i.e., early diagenesis in sediments, that in some regions is likely to affect simulated coastal ocean biogeochemistry of overlying waters. Moreover input of carbon and nutrients from rivers and groundwater is usually lacking. And even in models such as ours where that input is imposed as boundary conditions (Aumont et al., 2015), temporal variability and trends are neglected (Bauer et al., 2013; Cotrim da Cunha et al., 2007).

Nonetheless, coarse-resolution models are no longer the state of the art. Recently, there have been improvements in spatial resolution of global ocean models and the spatiotemporal resolution of surface forcing fields (Brodeau et al., 2010), thereby improving the representation of bathymetry and ocean processes in the highly variable coastal zone (Capet, 2004; Hofmann et al., 2011; McKiver et al., 2014). In any case, models currently provide the only means to estimate coastal uptake of anthropogenic carbon due to the lack of data-based estimates.

Here our aim is to estimate the air-to-sea flux of anthropogenic $CO_2$ into the coastal ocean and how it varies from region to region across the globe. We focus solely on the geochemical effect of anthropogenic $CO_2$ addition from the atmosphere to the ocean and neglect the role of varying river input and interactions with the sediment, as well as the feedback from a changing climate. To do so, we rely on an eddying version of the global NEMO circulation model (Madec, 2008), which also includes the LIM2 sea-ice model and is coupled to the PISCES biogeochemical model (Aumont and Bopp, 2006). More precisely, we use the ORCA05 eddy-admitting resolution, which ranges from $0.2°$ to $0.5°$ (i.e., 20 to 50 $km$). Although this resolution does not fully resolve coastal ocean bathymetry and dynamics, it does provide a first step into the eddying regime and a starting point upon which to compare future studies that will model the coastal ocean, globally, at higher resolution.

## 2  Methods

### 2.1  Coupled physical-biogeochemical model

For this study, we use version 3.2 of the ocean model known as NEMO (Nucleus for European Modelling of the Ocean), which includes (1) the primitive equation model Océan Parallélisé (OPA, Madec (2008)), (2) the dynamic-thermodynamic Louvain-La-Neuve sea-ice model (LIM, Fichefet and Morales Maqueda (1997)), and (3) the Tracer in the Ocean Paradigm (TOP), a passive tracer module that in this case is connected to version 1 of the ocean biogeochemical model PISCES (Pelagic Interaction Scheme for Carbon and Ecosystem Studies) (Aumont and Bopp, 2006). For the NEMO model, we use a global-scale configuration from the DRAKKAR community (see Barnier et al. (2006) and Timmermann et al. (2005)). Namely, we use the

ORCA05 global configuration, which possesses a curvilinear, tri-polar grid with a horizontal resolution that ranges between $0.2°$ near the North Pole to $0.5°$ at the equator (Fig. 1). Vertically, ORCA05 is discretized into 46 levels with thicknesses that range from $6\,\text{m}$ at the surface to $250\,\text{m}$ for the deepest ocean level (centered at $5625\,\text{m}$). Model bathymetry is computed from the 2' bathymetry file ETOPO2 from the National Geophysical Data Center. The numerical characteristics of our ORCA05 configuration follow the lead of Barnier et al. (2006) for the ORCA025 configuration with resolution-dependent modifications for the horizontal eddy diffusivity for tracers modified to $600\,\text{m}^2\,\text{s}^{-1}$ and horizontal eddy viscosity fixed to $-4 \times 10^{11}\,\text{m}^2\,\text{s}^{-1}$. To simulate the advective transport driven by geostrophic eddies, our ORCA05 simulation uses the eddy parameterization scheme of Gent and McWilliams (1990) applied with an eddy diffusion coefficient of $1000\,\text{m}^2\text{s}^{-1}$.

The biogeochemical model PISCES includes four plankton functional types: 2 phytoplankton (nanophytoplankton and diatoms) and 2 zooplankton (micro- and meso-zooplankton). PISCES also uses a mixed quota-Monod approach where (1) phytoplankton growth is limited by 5 nutrients (nitrate, ammonium, phosphate iron and silicate) following Monod (1949) and (2) elemental ratios of Fe, Si and Chl to C are prognostic variables based on the external concentrations of the limiting nutrients. In addition PISCES assumes a fixed C:N:P Redfield ratio set to $122 : 16 : 1$ from Takahashi et al. (1985) for both living and non-living pools. Similar to Geider et al. (1998), the phytoplankton Chl:C ratio in PISCES varies with photoadaptation. Furthermore, PISCES includes nonliving pools, namely a pool of semi-labile dissolved organic matter and two size classes of particulate organic matter. PISCES also explicitly accounts for biogenic silica and calcite particles. In PISCES, the sediment-water interface is treated as a reflective boundary condition where mass fluxes from particles are remineralized instantaneoulsy, except that small proportions of particle fluxes of organic matter, calcite, and biogenic silica escape the system through burial. Those burial rates are hence dependent on the local sinking fluxes, but are set to balance inputs from rivers and atmospheric deposition at the global scale. Thus global budgets of alkalinity and nutrients are balanced. For further details, we refer readers to Aumont and Bopp (2006).

To simulate carbon chemistry and air-sea $CO_2$ fluxes, the model follows the protocol from phase 2 of the Ocean-Carbon Cycle Model Intercomparison Project (OCMIP, Najjar and Orr (1999)) protocol. The sea-to-air $CO_2$ flux $FCO_2$ is computed using the following equation:

$$FCO_2 = \alpha\, k\, \Delta pCO_2 \tag{1}$$

where $\alpha$ is the solubility of $CO_2$ computed from Weiss (1974) and $\Delta pCO_2$ is the difference between the partial pressures of sea-surface and atmospheric $CO_2$. Thus $FCO_2$ is positive when $CO_2$ is transferred from the ocean to the atmosphere. The piston velocity $k$ is based on equation (3) of Wanninkhof (1992) as:

$$k = 0.30\, u_w^2 \sqrt{\frac{660}{Sc}}(1 - f_{ice}) \tag{2}$$

where $u_w$ is the wind speed at $10\,\text{m}$, $Sc$ is the $CO_2$ Schmidt number, and $f_{ice}$ is the ice fraction.

## 2.2 Simulations

The dynamic model was started from rest and spun up for 50 years. Initial conditions for temperature and salinity are as described by Barnier et al. (2006). Initial biogeochemical fields of nitrate, phosphate, oxygen and silicate are from the 2001 World Ocean Atlas (Conkright et al., 2002), whereas preindustrial dissolved inorganic carbon (DIC) and total alkalinity (Alk)

come from the GLODAP gridded product (Key et al., 2004). Conversely, because data for iron and dissolved organic carbon (DOC) are more limited, both those fields were initialized with model output from a 3000-year spin-up simulation of a global $2°$configuration of the same NEMO-PISCES model (Aumont and Bopp, 2006). All other biogeochemical tracers have much shorter time-scales; hence, they were initialized to globally uniform constants.

After the 50-year spin up, we launched 2 parallel simulations: the first was a historical simulation run during 1870 to 2012

(143 years), and forced with a spatially uniform and temporally increasing atmospheric mole fraction of $CO_2$ (from which PISCES computes atmospheric $pCO_2^{atm}$ following OCMIP2) reconstructed from ice-core and atmospheric records (Le Quéré et al., 2014); the second simulation is a parallel control run, where the 143-year simulation is identical except that it is forced with the preindustrial level of atmospheric mole fraction of $CO_2$ (287 ppm, constant in time). The preindustrial reference year is defined as 1870, thus neglecting changes in anthropogenic carbon storage in the ocean from 1750 to 1870. The $FCO_2$

computed with the historical simulation is for total carbon (total $FCO_2$), whereas that from the control simulation is for natural carbon (natural $FCO_2$). The corresponding anthropogenic $FCO_2$ is computed as the total minus natural $FCO_2$.

All simulations were forced identically, with atmospheric fields from the DRAKKAR Forcing Set (DFS, Brodeau et al. (2010)). These fields include zonal and meridional components of 10-m winds, 2-m air humidity, 2-m air temperature, downward shortwave and longwave radiation at the sea surface, and precipitation. More specifically the NEMO-PISCES model is

forced with version 4.2 of this forcing (DFS4.2, based on the ERA40 reanalysis) over 1958–2001, and that is followed by forcing from version 4.4 (DFS4.4) over 2002 to 2012. For the 1870–1957 period, where atmospheric reanalyses are unavailable, we repeatedly cycled the 1958-2007 DFS4.2 forcing.

Boundary conditions are also needed for biogeochemical tracers, i.e., besides the atmospheric-$CO_2$ connection mentioned already. The model's lateral input from river discharge of DIC and DOC are taken from the annual estimates of the Global

Erosion Model (Ludwig et al., 1996), constant in time. The DOC from river discharge is assumed to be labile and is directly converted to DIC upon its delivery to the ocean. Inputs of dissolved iron (Fe), nitrate ($NO_3^{2-}$), phosphate ($PO_4^{3-}$), and silicate ($SiO_2$) are computed from the sum of DIC and DOC river input using a constant set of ratios for C:N:P:Si:Fe, namely $320 : 16 : 1 : 53.3 : 3.64$ x $10^{-3}$, as computed from Meybeck (1982) for C:N, from Takahashi et al. (1985) for N:P, from de Baar and de Jong (2001) for Fe:C, and from Treguer et al. (1995) for Si:C. River discharge assumes no seasonal

variation. Atmospheric deposition of iron comes from Tegen and Fung (1995).

Here, we use the conventional definition of anthropogenic carbon in the ocean used by previous global-ocean model studies (OCMIP, http://ocmip5.ipsl.jussieu.fr/OCMIP/ and e.g., Bopp et al. (2015)), namely that anthropogenic carbon comes only from the direct geochemical effect of increasing atmospheric $CO_2$ and its subsequent invasion into the ocean. By definition,

this anthropogenic $FCO_2$ does not include any effect from potential changes in ocean physics or biology. In the model, there are no changes nor variability in riverine delivery of carbon and nutrients, and anthropogenic carbon is not buried in sediments.

Following the 50-year spin up and 143-year control simulation, the simulation remains far from equilibrium. Its global natural carbon flux is $-0.33 \pm 0.3 \, \mathrm{Pg\,C\,yr^{-1}}$ (corresponding to $CO_2$ uptake by the ocean) during the last 10 years of the control simulation (2003-2012), as compared to the estimate of natural carbon outgassing of $0.45 \, \mathrm{Pg\,C\,yr^{-1}}$ by Jacobson et al. (2007). That difference is partly due to the strategy for our simulations, which were initialized with data and spun up for only 50 years because of the computational constraints to make higher resolution simulations (ORCA05). At lower resolution (ORCA2), after a spin-up of 3000 years, there is $0.26 \, \mathrm{Pg\,C\,yr^{-1}}$ greater globally integrated sea-to-air flux, relative to results after only a 50-year spin up. Nearly all of that enhanced sea-to-air $CO_2$ flux due to the longer spin up comes from the Southern Ocean. Anthropogenic $FCO_2$ estimates are expected to be influenced very little by model drift because of the way anthropogenic carbon is defined, i.e., drift affects both natural carbon and total carbon in the same way.

## 2.3 Defining the global coastal ocean

To sample the global coastal ocean area, the model grid cells were selected following the Margins and Catchments Segmentation (MARCATS) of Laruelle et al. (2013), hereafter LA13. The outer limit of the coastal ocean is defined as the maximum slope at the shelf break, while the inner limit is taken as the coastline, thus excluding the proximal zone of the coastal ocean (Fig. 1). Hence, only the continental shelf area is taken into account. The MARCATS segmentation divides the global coastal ocean into 45 regional units (Table 2). The limits of each of theses units delineate areas that present roughly homogenous oceanic features such as coastal currents or the boundaries of marginal seas. Following the Liu et al. (2010) classification of the continental shelf seas, LA13 aggregated the 45 units into 7 classes with similar physical and oceanographic large-scale characteristics such as the Eastern Boundary Currents and the Polar Margins. The high-resolution Geographical Information System (GIS) file describing the MARCATS segmentation from LA13 was regridded using the QGIS software (QGIS Development Team, 2015) on the ORCA05 model grid in order to sample the model results on its own grid. This regridding technique implies some modifications to the regions initially described in LA13. In the model, the global coastal ocean has a total surface area of $27.0 \times 10^6 \, \mathrm{km^2}$, which is 8% less than the original value from Laruelle et al. (2014). Here, the model's total coastal ocean surface area represents 7.5% of the total area of the global ocean. Subsequently we refer to the individual MARCATS regions using the terminology of LA13.

## 2.4 Evaluation dataset

To evaluate the total $FCO_2$ simulated by the model (historical simulation), we compare it to the database from Laruelle et al. (2014), hereafter LA14, which provides observation-based estimates for that flux over the MARCATS regions. This database was constructed by aggregating $3 \times 10^6$ coastal sea-surface $pCO_2$ measurements collected during 1990 to 2011 and included in the Surface Ocean $CO_2$ Atlas version 2.0 (SOCAT v2.0, Pfeil et al. (2013); Bakker et al. (2014)). These measurements represent about 30% of the SOCAT v2.0 dataset. To compute the flux, LA14 also relied on wind speeds from the multiplatform CCMP wind-speed database (Atlas et al., 2011), atmospheric $CO_2$ from GLOBALVIEW-CO2 (2012), and the flux parameterization

from Wanninkhof (1992) as modified by Takahashi et al. (2009). As sensitivity tests, LA14 also used the flux parameterizations from Ho et al. (2006) as well as the original formulation from Wanninkhof (1992).

Thus LA14 computed mean annual $FCO_2$ estimates for 42 of the 45 MARCATS regions defined in LA13. The remaining MARCATS areas (12:Hudson Bay, 21:Black Sea and 29:Persian Gulf) are devoid of observations in the SOCAT database and were neglected. For the remaining regions, because of the large heterogeneity in both the spatial and temporal coverage of ocean $pCO_2$ observations, the uncertainties for each the MARCATS $FCO_2$ estimates from LA14 vary greatly. For example, only 28% of the sub-units of MARCATS regions used in LA14 have an estimate for $FCO_2$ uncertainty of less than $0.25 \, \mathrm{mol \, C \, m^{-2} \, yr^{-1}}$. The data-based $FCO_2$ estimate for the Sea of Okhotsk is not taken into account due to the extremely poor data coverage of this region and its strong divergence with the local literature (LA14). Here, we do not evaluate the simulated annual cycle of flux of total carbon because few MARCATS regions provide adequate temporal coverage. Finally, LA14 is the first and only study to provide coastal-ocean observation-based $FCO_2$ estimates at global scale taking into account for the reduction in $FCO_2$ due to sea-ice cover along coasts; hence it is directly comparable to our model results.

Besides the coastal data-based estimates of $FCO_2$ from LA14, we also compare our model results to those for the open ocean from Takahashi et al. (2009) and Landschützer et al. (2014). Both the global and coastal observational estimates are compared to the average modeled $FCO_2$ over the last 20 years (1993–2012) of the historical simulation. For the coastal comparison, simulated total $FCO_2$ are spatially averaged over each MARCATS regions. In addition, the model's uncertainty, computed as the interannual variability over 1993–2012, is compared to uncertainties in the observational estimates, computed as the standard deviation between flux parameterizations from Wanninkhof (1992) as modified by Takahashi et al. (2009), Ho et al. (2006) and Wanninkhof (1992).

## 2.5 Revelle factor calculation

To assess how the capacity of the coastal ocean to absorb anthropogenic carbon differs from open-ocean surface waters, we computed the Revelle factor ($R_f$, Sundquist et al. (1979)) using the CO2SYS MATLAB algorithm (Van Heuven et al., 2011). CO2SYS was used using the simulated sea-surface temperature, salinity, alkalinity, and DIC for model years 1993–2012 with the total pH scale, the $K_1$ and $K_2$ constants from Lueker et al. (2000), the $K_{SO_4}$ constant from Dickson (1990) and the formulation of the borate-to-salinity ratio from Uppström (1974).

## 2.6 Residence time

To compute water residence time in each MARCATS region, we divided the volume of each region by the integrated outflow of water from 5-day mean current velocities at coastal boundaries from 2011.

## 3 Results

### 3.1 Global ocean fluxes

The simulated global-ocean uptake of anthropogenic carbon increases roughly linearly from 1950 to 2012, reaching an average of $2.3 \, \mathrm{Pg\,C\,yr^{-1}}$ during 1993–2012. That is comparable to the estimate from the fifth assessment report of the IPCC (Ciais et al., 2013) of $2.3 \pm 0.7 \, \mathrm{Pg\,C\,yr^{-1}}$ for 2000–2009 (Fig. 2).

Regionally, overall patterns in the total FCO$_2$ are similar between the model and data-based estimates from Landschützer et al. (2014) and Takahashi et al. (2009) (Fig. 3). Carbon is lost from the ocean in the equatorial band and in coastal upwelling regions, while it is gained by the ocean in the northern high latitudes. Quantitative comparison of the annual-mean map from the model with that from the Takahashi et al. (2009) observation-based database gives a root mean square error (RMSE) of $0.73 \, \mathrm{mol\,C\,m^{-2}\,yr^{-1}}$ and a correlation coefficient R of 0.80; likewise, comparison with the Landschützer et al. (2014) observational-based database gives a similar RMSE ($0.70 \, \mathrm{mol\,C\,m^{-2}\,yr^{-1}}$) and R (0.81). Integrating over latitudinal bands, (Table 1), the model overestimates carbon uptake for the 90°S-30°S region where it absorbs $1.50 \, \mathrm{Pg\,C\,yr^{-1}}$ of total carbon versus $0.73\text{-}0.77 \, \mathrm{Pg\,C\,yr^{-1}}$ from Takahashi et al. (2009) and Landschützer et al. (2014) observational databases. This may be a signature of the fact that the model simulation is still far from equilibrium (see section 2.2 paragraph 5 for details). The model also underestimates outgassing in the tropical band, where it releases $0.13 \, \mathrm{Pg\,C\,yr^{-1}}$ vs. $0.13\text{-}0.20 \, \mathrm{Pg\,C\,yr^{-1}}$ for the 2 data-based estimates. Further north in the 30°N-90°N band the model takes up $0.93 \, \mathrm{Pg\,C\,yr^{-1}}$ vs. $1.53\text{–}1.59 \, \mathrm{Pg\,C\,yr^{-1}}$ for Takahashi et al. (2009) and Landschützer et al. (2014).

### 3.2 Coastal ocean fluxes

#### 3.2.1 Total CO$_2$

The simulated uptake of total carbon by the coastal ocean averages $267 \, \mathrm{Tg\,C\,yr^{-1}}$ during the 1993–2012. Most of the 45 MARCATS regions act as carbon sinks; together, they absorb $283 \, \mathrm{Tg\,C\,yr^{-1}}$. The largest uptake is $3.4 \, \mathrm{mol\,C\,m^{-2}\,yr^{-1}}$ in the South Greenland region. Few MARCATS regions act as carbon sources to the atmosphere (Table 2 and Fig. 4.a), i.e., only 14% of the global coastal-ocean surface area, together losing $16 \, \mathrm{Tg\,C}$ of carbon to the atmosphere every year. The mean annual carbon loss per square meter in these MARCATS regions is usually relatively weak, less than $1.5 \, \mathrm{mol\,C\,m^{-2}\,yr^{-1}}$). When grouped into MARCATS classes (see Table 3), all classes are carbon sinks, absorbing from 0.06 to $1.65 \, \mathrm{mol\,C\,m^{-2}\,yr^{-1}}$. By class, the largest specific fluxes occur in the Western Boundary Current regions and the Subpolar Margins, which absorb 1.65 and $1.61 \, \mathrm{mol\,C\,m^{-2}\,yr^{-1}}$, respectively. More generally, the tropical MARCATS regions act as weak carbon sources and the mid-to-high latitude regions act as strong carbon sinks (Fig. 4.a). The same trend is also apparent in the zonal-mean distribution (Fig. 5).

A comparison of the simulated vs. observed FCO$_2$ estimates for each MARCATS region is reported in Table 2 and on Fig. 6. The Pearson correlation coefficient R is 0.8 for specific fluxes. In the model, 79% of the MARCATS regions act as carbon sinks, whereas that proportion is 64% for LA14. After aggregating the specific flux estimates into the different MARCATS

classes (Table 3 and Fig. 7), the correlation coefficient R increases to 0.9. Generally, our model results tend to simulate larger sinks and weaker sources than observed (i.e. 76% of the specific simulated fluxes of total carbon have lower relative values than the data-based estimates). For some MARCATS classes, even the sign of the simulated flux differs from the data-based estimates, e.g., for the Indian Margins and the Eastern Boundary Currents. The latter class contains two regions (Moroccan and S-W Africa Upwelling) having the worst overall agreement. Otherwise, in the Arctic polar regions, the simulated uptake is too low, with 52 $\mathrm{Tg\,C\,yr}^{-1}$ from the model vs. 86 $\mathrm{Tg\,C\,yr}^{-1}$ from LA14.

### 3.2.2 Anthropogenic $CO_2$

The anthropogenic $FCO_2$ is computed as the difference between the total flux (historical simulation) and natural flux (control simulation). When integrated over the global coastal ocean, the mean anthropogenic flux during 1993–2012 is $0.10 \pm 0.01\ \mathrm{Pg\,C\,yr}^{-1}$. That amounts to 4.5% of the simulated global anthropogenic carbon uptake, substantially less than the 7.5% proportion of the coastal-to-global ocean surface areas. During 1950–2000, the uptake of anthropogenic carbon by the coastal ocean grows essentially linearly as it does for the global ocean. That is, it grows at a nearly constant rate of $0.0015\ \mathrm{Pg\,C\,yr}^{-2}$, which is 4.4% of the rate for the global ocean increase in anthropogenic carbon uptake over the same period (Fig. 2).

All MARCATS regions absorb anthropogenic carbon at rates ranging from 0.01 $\mathrm{mol\,C\,m}^{-2}\,\mathrm{yr}^{-1}$ for the Baltic Sea to 0.86 $\mathrm{mol\,C\,m}^{-2}\,\mathrm{yr}^{-1}$ for the South Greenland region (Table 2 and Fig. 4.b). By class, the strongest specific fluxes of anthropogenic carbon into the ocean occur in the boundary current regions, namely the EBC and WBC, with 0.42 and 0.48 $\mathrm{mol\,C\,m}^{-2}\,\mathrm{yr}^{-1}$, respectively. Conversely, the weakest anthropogenic carbon uptake occurs in the Tropical Margins and the Indian margins with 0.22 and 0.24 $\mathrm{mol\,C\,m}^{-2}\,\mathrm{yr}^{-1}$, respectively. But specific fluxes can be misleading. Although the Polar and Subpolar margins do not have the highest specific fluxes, their integrated uptake of anthropogenic carbon is large because of their large surface areas (Fig. 4.b and 4.c). Together they absorb 46 $\mathrm{Tg\,C\,yr}^{-1}$, which is 45% of total uptake of anthropogenic carbon by the global coastal ocean.

These results emphasize that there is no link between anthropogenic and total carbon fluxes when comparing patterns between regions. For example, even though the EBC and WBC regions are the most efficient regions in anthropogenic carbon uptake (both above 0.4 $\mathrm{mol\,C\,m}^{-2}\,\mathrm{yr}^{-1}$), their behavior differs greatly in terms of the flux of total carbon, i.e., -1.65 versus -0.12 $\mathrm{mol\,C\,m}^{-2}\,\mathrm{yr}^{-1}$, respectively (Fig. 7). The same lack of correlation between anthropogenic and total flux patterns is even clearer in the zonal mean distributions (Fig. 5). For instance, the specific fluxes of anthropogenic carbon into the coastal ocean between 55°S and 90°N are nearly uniform, remaining near -0.5 $\mathrm{mol\,C\,m}^{-2}\,\mathrm{yr}^{-1}$; conversely, the total carbon fluxes vary greatly, between -2 to +0.5 $\mathrm{mol\,C\,m}^{-2}\,\mathrm{yr}^{-1}$. Theses variations in the total carbon flux are dictated by variations in the natural carbon flux (Fig. 5).

## 4 Discussion

### 4.1 Comparison with previous coastal estimates

#### 4.1.1 Total flux

Our mean simulated uptake of total carbon by the global coastal ocean during the 1993–2012 is $0.27 \pm 0.07 \, \mathrm{Pg \, C \, yr^{-1}}$, which
falls within the range of previous data-based estimates of $0.2$–$0.4 \, \mathrm{Pg \, C \, yr^{-1}}$ (Borges et al., 2005; Cai et al., 2006; Chen and
Borges, 2009; Laruelle et al., 2010; Cai, 2011; Chen et al., 2013; Laruelle et al., 2014). Out of those, estimates provided since
2011 gather closer to the lower limit, e.g., the estimate of $0.2 \, \mathrm{Pg \, C \, yr^{-1}}$ from LA14, as is also the case for our model-based
estimate. Some aspects of the LA14 data-based approach are shared by our model-based approach, i.e., the same reference
period, essentially the same definition of the coastal ocean and the same correction for the effect of sea-ice cover on $FCO_2$.
LA14 is the first observation-based study to take into account this sea-ice effect for coastal-ocean $FCO_2$ estimates at global
scale.

Using a box model, Andersson and Mackenzie (2004) and Mackenzie et al. (2004) estimated the global coastal ocean acted
as a carbon source to the atmosphere prior to industrialisation; however, they also estimate that industrialisation has recently
led to a reversal in the sign of this flux (the global coastal ocean became a carbon sink) mainly due to the enhancement of NEP
from increased riverine inputs. In contrast, our model simulations indicate that the preindustrial coastal ocean was already a
carbon sink, and that that sink has strengthened over the industrial period. This discrepancy appears to be explained by different
definitions of the coastal ocean. Both the box model and our 3-D model include the distal coastal zone, but only the box model
includes the proximal coastal zone (bays, estuaries, deltas, lagoons, salt marshes, mangroves, and banks). That proximal zone
is known generally as a strong source of carbon to the atmosphere (Rabouille et al., 2001).

The model representation of riverine DOC input and its instantaneous remineralization has potential implications for our
estimates of total $FCO_2$. In the Amazon plume for instance, we underestimate $CO_2$ absorption because of this instantaneous
addition of DIC without input of alkalinity. However this assumption has no direct implication on our anthropogenic $FCO_2$
estimates.

Furthermore, our simplified representation of sedimentary processes affects simulated total $CO_2$ fluxes (Krumins et al.,
2013; Soetaert et al., 2000). First, the model lacks an explicit representation of sedimentary processes. Thus it cannot reproduce
the temporal dynamics of interactions between sediments and the overlying water column, e.g., resulting in potential delays
between sediment burial and remineralization. Second, our model neglects any alkalinity source from sediment anaerobic
degradation, such as denitrification and sulfate reduction of deposited organic matter. Even if not well constrained (Chen,
2002; Thomas et al., 2009; Hu and Cai, 2011; Krumins et al., 2013), this source of alkalinity could partially balance the total
$CO_2$ uptake of the coastal ocean. However, the simplified representation of these sediment processes has no direct effect on
our anthropogenic $FCO_2$ estimates.

### 4.1.2 Anthropogenic flux

The strongest specific fluxes of anthropogenic carbon into the ocean occur in the boundary current regions, namely the EBC and WBC. Indeed, these regions show significant vertical and lateral mixing features such as filaments and eddies from the strong adjacent western boundary currents and upwelling from Eastern Boundary Upwelling Systems (EBUS). Those physical processes lead to deepen the mixed layer depth, export the absorbed anthropogenic carbon from shallow water to deeper water layers and export it to the adjacent open ocean.

Our estimate of the simulated anthropogenic carbon uptake of $0.10\,\mathrm{Pg\,C\,yr^{-1}}$ for the global coastal ocean (Fig. 9) is about half that found by Wanninkhof et al. (2013) for a similar period. The latter study estimates coastal anthropogenic $CO_2$ uptake by extrapolating specific $FCO_2$ from the adjacent open ocean into coastal areas, exploiting coarse-resolution models and data. To compare approaches, we applied the Wanninkhof et al. (2013) extrapolation method to our model output; we found the same result as theirs for global coastal ocean uptake of anthropogenic $CO_2$ ($0.18\,\mathrm{Pg\,C\,yr^{-1}}$). Thus the extrapolation technique leads to an overestimate of anthropogenic $CO_2$ uptake in the model's global coastal ocean.

Nonetheless, the Wanninkhof et al. (2013) estimate for the anthropogenic carbon uptake by the coastal ocean was used by Regnier et al. (2013) for their coastal carbon budget. That budget also accounts for the increase in river discharge of carbon ($0.1\,\mathrm{Pg\,C\,yr^{-1}}$) and nutrients during the industrial era, which promotes organic carbon production, some of which is buried in the coastal zone (up to $0.15\,\mathrm{Pg\,C\,yr^{-1}}$). Unfortunately, these numbers remain particularly uncertain. Hence we have chosen to ignore them, adopting the conventional definition of anthropogenic carbon in the ocean used by previous global-ocean model studies, namely that anthropogenic carbon comes only from the direct geochemical effect of the anthropogenic increase in atmospheric $CO_2$ and its subsequent invasion into the ocean. The future challenge of improving estimates of changes and variability in riverine delivery of carbon and nutrient and sediment burial is critical to refine land contributions to the coastal ocean carbon budget.

Our estimate of $0.10\,\mathrm{Pg\,C\,yr^{-1}}$ for the anthropogenic $FCO_2$ into the coastal ocean is 40% less than the $0.17\,\mathrm{Pg\,C\,yr^{-1}}$ estimated by Borges (2005) from Andersson and Mackenzie (2004) and Mackenzie et al. (2004). Causes for this difference may stem from (1) the different definitions of the coastal ocean (proximal coastal zone included in the box model but not the 3-D model), (2) the different approaches (uniform coastal ocean in the box model but not in the 3-D model), and (3) the role of sediments (pore waters included in the box model but neglected in the 3-D model).

### 4.2 Coastal vs. open ocean

Patterns in our simulated total $FCO_2$ in the coastal ocean generally follow those for the open ocean, with net carbon sources in the low latitudes and carbon sinks in the mid- to high-latitudes (Fig. 5). The same tendency was pointed out by Gruber (2014) when discussing the LA14 data-based fluxes. The patterns in our simulated total $CO_2$ flux are mainly driven by patterns in the natural $CO_2$ flux both in the coastal and open oceans (Fig. 5). Yet the pattern for anthropogenic $CO_2$ flux differs greatly from that of natural $CO_2$, having its strongest uptake in the Southern Ocean in both the open and coastal oceans, i.e., where zonally averaged specific uptake reaches up to $1.5\,\mathrm{mol\,C\,m^{-2}\,yr^{-1}}$. The bathymetry of MARCATS regions around the Antarctic

continent is much deeper than in the other coastal regions (500 m vs. 160 m for the global coastal ocean); this probably reduces the contrast between the coastal and open ocean in the Southern Ocean and explains the similarities of anthropogenic carbon uptake rates there.

Despite large-scale similarities between coastal and open-ocean fluxes of total carbon, some coastal regions differ substan-
tially from those in the adjacent open ocean waters (Fig. 3.a). These local differences are particularly apparent around coastal upwelling systems, i.e., in the Western Arabian Sea and in Eastern Boundary Upwelling Systems (EBUS), such as the Peru-vian Upwelling Current, the Moroccan Upwelling, and the Southern Western Africa upwelling. Some of these coastal regions act as strong total carbon sources, with mean carbon fluxes of up to $1.44 \, \mathrm{mol \, C \, m^{-2} \, yr^{-1}}$, whereas surrounding open-ocean waters exhibit little FCO$_2$ (fluxes close to $0 \, \mathrm{mol \, C \, m^{-2} \, yr^{-1}}$). Other regions also exhibit large coastal-open ocean contrasts,
including the Tropical Western Atlantic where there is a massive loss of carbon at the location of the Amazon river discharge. However the carbon sink in the Amazon river plume reported in Lefèvre et al. (2010) is not reproduced. This discrepancy may be due to the modelled instantaneous remineralisation of land-derived DOC or to shortcomings in the model representation of sedimentary processes.

A key finding of our model study is that the flux of anthropogenic CO$_2$ into the coastal ocean ($0.10 \, \mathrm{Pg \, C \, yr^{-1}}$) is half the
previous estimate (Wanninkhof et al., 2013). Unlike in that study, our specific flux of anthropogenic CO$_2$ is substantially lower for the global coastal ocean than for the global open ocean (i.e., -0.31 vs. $-0.54 \, \mathrm{mol \, C \, m^{-2} \, yr^{-1}}$ for the 1993–2012 average). Although the coastal ocean surface area is 7.5% that of the global ocean, it absorbs only 4.5% of the globally integrated flux of anthropogenic carbon into the ocean.

Our estimate for coastal ocean uptake of anthropogenic carbon is ten times smaller than the $1 \, \mathrm{Pg \, C \, yr^{-1}}$ estimate by Tsuno-
gai et al. (1999) associated with his proposed continental shelf pump (CSP). However, Tsunogai's CSP is based on contempo-rary measurements and thus concerns total carbon, not the anthropogenic change. That nuance is critical because contemporary estimates of fluxes are not directly comparable to anthropogenic fluxes nor global budgets of carbon from the IPCC and the Global Carbon Project, both focused on the anthropogenic change. Unfortunately Tsunogai et al. (1999) prompted confusion by stating that their total carbon flux into the coastal ocean was equivalent to half of the global-ocean uptake of anthropogenic
carbon. The same confusion prompted Thomas et al. (2004) to emphasize that the coastal ocean contributes more to the global carbon budget than expected from its surface area.

The lower specific flux of anthropogenic CO$_2$ into the global coastal ocean relative to the average for the open ocean could have 2 causes: (1) physical factors, e.g., if vertical mixing in the coastal ocean is relatively weak or if there is a bottleneck in the offshore transport carbon and (2) chemical factors, if coastal waters have a lower chemical capacity to absorb anthropogenic
carbon (lower carbonate ion concentration, higher Revelle factor $R_f$).

To assess how $R_f$ differs between coastal and open-ocean surface waters, we computed it using CO2SYS from simulated sea-surface temperature, salinity, alkalinity, and DIC for model years 1993–2012. Thus we computed mean Revelle factors of 12.5 for the global coastal ocean, 10.9 for the global ocean, 9.2 for the tropical oceans (30°S-30°N), and 12.8 for the Southern Ocean (90°S-30°S). And these tendencies are persistent. During 1910–2012, the average coastal-ocean Revelle factor remains
15% larger than that for the open ocean. Hence average surface waters in the model's coastal ocean have a lower chemical

capacity to take up anthropogenic carbon than do average surface waters of the global ocean. That finding is consistent with the lower simulated specific fluxes of anthropogenic carbon into the coastal ocean. Yet it is not only the chemical capacity that matters. For example, despite similar chemical capacities, the specific flux of anthropogenic carbon into Southern Ocean is more than twice that of the global coastal ocean. Thus we must turn to physical factors to help explain the lower efficiency of

the coastal ocean to take up anthropogenic carbon.

Out of the $0.10 \, \mathrm{Pg\,C\,yr^{-1}}$ absorbed by the coastal ocean, we find that only 70% (i.e. $0.07 \, \mathrm{Pg\,C\,yr^{-1}}$) is transferred to the open ocean (Fig. 9). Thus $0.03 \, \mathrm{Pg\,C\,yr^{-1}}$ of anthropogenic carbon accumulates in the coastal-ocean water column during 1993–2012. That simulated accumulation is not significantly different from the estimate of $0.05 \pm 0.05 \, \mathrm{Pg\,C\,yr^{-1}}$ from Regnier et al. (2013). The accumulation in the coastal ocean is effective over the entire period (1910-2012) as the uptake of

anthropogenic carbon by the global coastal ocean is always inferior to its cross-shelf export (Fig. 10). To gain insight into this cross-shelf exchange, we computed the simulated mean water residence times for each MARCATS region (Fig. 8). Residence times for most coastal regions are of the order of a few months or less, except for Hudson Bay, the Baltic Sea and the Persian Gulf. The latter three regions are generally more confined and we expect longer residence times, although our model simulations were never designed to simulate these regions accurately. Generally, our simulated residence times are shorter than what

has been published for similarly defined coastal regions although methods differ substantially (Jickells, 1998; Men and Liu, 2014; Delhez et al., 2004). Despite these short residence times, the cross-shelf export of anthropogenic carbon is unable to keep up with the increasing air-sea flux of anthropogenic carbon (Fig. 10). This may be explained by the open-ocean waters that are imported to the coastal ocean being already charged with anthropogenic carbon, thus limiting further uptake in the coastal zone. This accumulation rate of anthropogenic carbon in the coastal ocean contrasts with the lower simulated propor-

tion that remains in the mixed layer of the global ocean. Using a coarse-resolution global model, Bopp et al. (2015) showed that on average for the global ocean, only ∼10% of the anthropogenic carbon that crosses the air-sea interface accumulates in the seasonally-varying mixed layer. The CSP hypothesis from Tsunogai et al. (1999) assumes that much of the $1 \, \mathrm{Pg\,C\,yr^{-1}}$ of total carbon absorbed by the coastal ocean is exported to the deep ocean. Also assuming that the CSP operates equally in all shelf regions across the world, Yool and Fasham (2001) used coarse- resolution global model to estimate that 53% of the coastal

uptake is exported to the open ocean. Yet they considered only natural carbon. Conversely, we focus purely on anthropogenic carbon. Our simulations suggest that 70% of the anthropogenic carbon absorbed by the coastal ocean during 1993–2012 is transported offshore to the deeper open ocean.

## 5   Conclusions

The goal of this study is to estimate the anthropogenic $CO_2$ flux from the atmosphere to the coastal ocean, both globally

and regionally, using an eddying global-ocean model, making 143-year simulations forced by atmospheric reanalysis data and atmospheric $CO_2$. We first evaluate the simulated air-sea fluxes of total $CO_2$ for 45 coastal regions and find a correlation coefficient R of 0.8 when compared to observation-based estimates. Then we estimate the average simulated anthropogenic carbon uptake by the global coastal ocean over 1993–2012 to $0.10 \pm 0.01 \, \mathrm{Pg\,C\,yr^{-1}}$, equivalent to 4.5% of global-ocean

uptake of anthropogenic $CO_2$, an amount less than expected based on the surface area of the global coastal ocean (7.5% of the global ocean). Furthermore, our estimate is only about half of that estimated by Wanninkhof et al. (2013), whose budget was based on extrapolating adjacent open-ocean data-based estimates of the specific flux into the coastal ocean. We attribute our lower specific flux of anthropogenic carbon into the global coastal ocean mainly to the model's associated offshore carbon transport, which is not strong enough to reduce surface levels of anthropogenic DIC (and thus anthropogenic $p$CO$_2$) to levels that are as low as those in the open ocean (on average). Whether or not our model provides a realistic estimate of offshore transport at the global scale is a critical question, however, that demands further investigation.

Clearly, our approach is limited by the extent to which the coastal ocean is resolved. Our model's horizontal resolution does not allow it to fully resolve some fine-scale coastal processes such as tides, which affect FCO$_2$ at tidal fronts (Bianchi et al., 2005). Model resolution is also inadequate to fully resolve mesoscale and sub-mesoscale eddies and associated upwelling. Moreover, in the mid-latitudes with a water depth of 80 m, the first baroclinic Rossby radius (the dominant scale affecting coastal processes) is around 200 km but it falls below 10 km on Arctic shelves (Holt et al., 2014; Nurser and Bacon, 2014). Thus the higher latitudes need much finer resolution (Holt et al., 2009).

Yet all model studies must weigh the costs and benefits of pushing the limits toward improved realism. Our approach has been to use a model that takes only a first step into the eddying regime in order to be able to achieve long physical-biogeochemical simulations with atmospheric $CO_2$ increasing from preindustrial levels to today. It represents a step forward when compared to studies with typical coarse resolution ocean models (around 2° horizontal resolution), which may be considered to be designed exclusively for the open ocean. In the coming years, increasing computational resources will allow further increases in spatial resolution and a better representation of the coastal ocean in global ocean carbon cycle models.

Improvements will also be needed in terms of the modeled biogeochemistry of the coastal zone. Most global-scale biogeochemical models neglect river input of nutrients and carbon. Although that is taken into account in our simulations, the river input forcing is constant in time (Aumont et al., 2015). Seasonal and higher frequency variability in carbon and nutrient river input (e.g., from floods and droughts) is substantial as often are anthropogenic trends. For simplicity, virtually all global-scale models neglect sediment resuspension and early diagenesis in the coastal-zone. Those processes in some coastal areas may well alter nutrient availability, surface DIC, and total alkalinity, which would affect FCO$_2$. In addition, in the coastal zone, one must eventually go beyond the classic definition of anthropogenic carbon, i.e., the change due only to the direct influence of the anthropogenic increase in atmospheric $CO_2$ on the FCO$_2$ and ocean carbonate chemistry. Changes in other human induced perturbations may be substantial. For example, an important research topic will be to better assess potential changes in sediment burial of carbon in the coastal zone during the industrial era, estimated at up to $0.15\,\mathrm{Pg\,C\,yr^{-1}}$ but with large uncertainty (Regnier et al., 2013).

To improve understanding of the critical land-ocean connection and its role in carbon and nutrient exchange, we call for a long-term effort to exploit the latest, global-scale, high-resolution, ocean general circulation models, adding ocean biogeochemistry, and improving them to better represent the coastal and open oceans together as one seamless system.

*Acknowledgements.* We are grateful to the associate editor J. Middelburg for managing the peer-reviewing process and to two referees, N. Gruber and P. Regnier, for their helpful comments. We thank G. G. Laruelle for providing the raw data version of the Laruelle et al. (2014) database, and we are grateful to C. Nangini and P. Brockmann for their help with the GIS analysis, data treatment, and data visualization. We also thank J. Simeon for the first implementation of the ORCA05-PISCES model. The research leading to these results was supported

5    through the EU FP7 project CARBOCHANGE (grant 264879), the EU H2020 project CRESCENDO project (grant 641816), and the EU H2020 project C-CASCADES (Marie Sklodowska-Curie grant 643052). Simulations were made using HPC resources from GENCI-IDRIS (grant x2015010040). TB was funded through a scholarship cosupported by DGA-MRIS and CEA scholarship.

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

**Table 1.** Sea-to-air total $CO_2$ fluxes $(\mathrm{Pg\,C\,yr^{-1}})$ given as zonal means from Takahashi et al. (2009) for the reference year 2000, from Landschützer et al. (2014) for 1998-2011 and from the ORCA05 model for 1993–2012.

| Latitudinal bands | Observation-based climatologies | | This study |
|---|---|---|---|
| | Takahashi et al. (2009) | Landschützer et al. (2014) | ORCA05 |
| 90°S - 30°S | -0.77 | -0.73 | -1.50 |
| 30°S - 30°N | 0.20 | 0.13 | 0.13 |
| 30°N - 90°N | -1.59 | -1.53 | -0.93 |

**Table 2.** MARCATS regions as described by Laruelle et al. (2013, 2014), along with means for data-based fluxes of total $CO_2$ from LA14 during 1990-2011 as well as simulated anthropogenic and total $CO_2$ fluxes, and residence time during 1993–2012. Uncertainties are the interannual variability over the averaged period. Abbreviations are included for North (N), South (S), East (E), West (W), Eastern Boundary Current (EBC); Western Boundary Current (WBC), sea-to-air flux of total carbon ($FCO_2^{tot}$), anthropogenic carbon ($FCO_2^{ant}$). Surface areas indicated as 'from LA14' actually differ slightly from those published in LA13 as they have been modified for subsequent computations (Goulven G. Laruelle, personal communication, January 2015).

| N° | System Name | Class | Surface ($10^3$ km$^2$) Model | LA14 | $FCO_2^{tot}$ (mol C m$^{-2}$ yr$^{-1}$) Simulated | LA14 | $FCO_2^{tot}$ (Tg C yr$^{-1}$) Simulated | LA14 | $FCO_2^{ant}$ mol C m$^{-2}$ yr$^{-1}$ | Tg C yr$^{-1}$ | Residence time (month) |
|---|---|---|---|---|---|---|---|---|---|---|---|
| 1 | N-E Pacific | Subpolar | 397 | 350 | -2.29 ± 0.17 | -1.61 | -10.935 ± 0.823 | -6.775 | -0.45 ± 0.05 | -2.16 ± 0.23 | 0.83 ± 0.23 |
| 2 | Californian Current | EBC | 118 | 208 | -0.34 ± 0.10 | -0.05 | -0.477 ± 0.148 | -0.135 | -0.35 ± 0.09 | -0.50 ± 0.13 | 1.00 ± 0.23 |
| 3 | Tropical E Pacific | Tropical | 152 | 183 | -0.12 ± 0.05 | 0.09 | -0.222 ± 0.095 | 0.192 | -0.36 ± 0.05 | -0.65 ± 0.10 | 0.51 ± 0.09 |
| 4 | Peruvian Upwelling Current | EBC | 138 | 143 | 1.44 ± 0.80 | 0.65 | 2.386 ± 1.325 | 1.073 | -0.39 ± 0.09 | -0.64 ± 0.15 | 0.72 ± 0.15 |
| 5 | Southern America | Subpolar | 1126 | 1190 | -1.51 ± 0.13 | -1.31 | -20.460 ± 1.705 | -18.715 | -0.46 ± 0.05 | -6.28 ± 0.74 | 0.65 ± 0.05 |
| 6 | Brazilian Current | WBC | 475 | 484 | -0.33 ± 0.08 | 0.10 | -1.872 ± 0.479 | 0.567 | -0.34 ± 0.05 | -1.95 ± 0.29 | 0.26 ± 0.06 |
| 7 | Tropical W Atlantic | Tropical | 479 | 488 | 0.86 ± 0.10 | 0.07 | 4.934 ± 0.551 | 0.394 | -0.26 ± 0.05 | -1.50 ± 0.31 | 0.20 ± 0.02 |
| 8 | Caribbean Sea | Tropical | 303 | 358 | 0.10 ± 0.10 | 0.81 | 0.366 ± 0.348 | 3.460 | -0.31 ± 0.04 | -1.12 ± 0.14 | 0.32 ± 0.03 |
| 9 | Gulf of Mexico | Marginal Sea | 469 | 532 | -0.79 ± 0.11 | -0.33 | -4.478 ± 0.633 | -2.100 | -0.32 ± 0.03 | -1.81 ± 0.16 | 1.01 ± 0.15 |
| 10 | Florida Upwelling | WBC | 545 | 591 | -2.25 ± 0.21 | -0.38 | -14.692 ± 1.351 | -2.723 | -0.66 ± 0.06 | -4.29 ± 0.36 | 0.39 ± 0.02 |
| 11 | Sea of Labrador | Subpolar | 576 | 638 | -1.27 ± 0.18 | -1.72 | -8.808 ± 1.244 | -13.172 | -0.32 ± 0.03 | -2.19 ± 0.21 | 1.20 ± 0.35 |
| 12 | Hudson Bay | Marginal Sea | 998 | 1064 | 0.31 ± 0.29 | n.d | 3.757 ± 3.423 | n.d. | -0.08 ± 0.04 | -0.99 ± 0.46 | 51.22 ± 22.75 |
| 13 | Canadian Archipelago | Polar | 1001 | 1145 | -0.52 ± 0.06 | -1.02 | -6.234 ± 0.748 | -13.986 | -0.09 ± 0.02 | -1.03 ± 0.21 | 2.82 ± 0.46 |
| 14 | N Greenland | Polar | 544 | 602 | -0.97 ± 0.15 | -0.61 | -6.333 ± 1.000 | -4.400 | -0.26 ± 0.05 | -1.67 ± 0.33 | 2.38 ± 0.44 |
| 15 | S Greenland | Polar | 238 | 262 | -3.35 ± 0.44 | -3.81 | -9.564 ± 1.259 | -11.972 | -0.86 ± 0.19 | -2.45 ± 0.53 | 0.48 ± 0.09 |
| 16 | Norwegian Basin | Polar | 141 | 162 | -2.87 ± 0.23 | -1.72 | -4.855 ± 0.396 | -3.342 | -0.60 ± 0.09 | -1.02 ± 0.15 | 0.31 ± 0.10 |
| 17 | N-E Atlantic | Subpolar | 1020 | 1073 | -2.16 ± 0.12 | -1.33 | -26.501 ± 1.419 | -17.165 | -0.53 ± 0.05 | -6.52 ± 0.59 | 0.93 ± 0.11 |
| 18 | Baltic Sea | Marginal Sea | 324 | 364 | 0.30 ± 0.07 | 0.51 | 1.184 ± 0.288 | 2.245 | -0.01 ± 0.01 | -0.05 ± 0.03 | 17.37 ± 9.52 |
| 19 | Iberian Upwelling | EBC | 251 | 267 | -1.13 ± 0.12 | 0.04 | -3.393 ± 0.352 | 0.122 | -0.27 ± 0.03 | -0.82 ± 0.10 | 2.31 ± 0.54 |
| 20 | Mediterranean Sea | Marginal Sea | 423 | 529 | -0.24 ± 0.06 | 0.62 | -1.196 ± 0.327 | 3.925 | -0.30 ± 0.02 | -1.52 ± 0.12 | 0.72 ± 0.09 |
| 21 | Black Sea | Marginal Sea | 131 | 172 | -0.24 ± 0.11 | n.d. | -0.375 ± 0.174 | n.d. | -0.18 ± 0.02 | -0.28 ± 0.03 | 1.60 ± 0.48 |
| 22 | Moroccan Upwelling | EBC | 177 | 206 | 0.18 ± 0.12 | 2.92 | 0.385 ± 0.263 | 7.220 | -0.33 ± 0.03 | -0.71 ± 0.07 | 0.67 ± 0.14 |
| 23 | Tropical E Atlantic | Tropical | 225 | 259 | 0.09 ± 0.08 | -0.06 | 0.239 ± 0.208 | -0.174 | -0.19 ± 0.02 | -0.52 ± 0.05 | 0.59 ± 0.09 |
| 24 | S W Africa | EBC | 300 | 298 | 0.43 ± 0.40 | -1.43 | 1.544 ± 1.448 | -5.103 | -0.59 ± 0.08 | -2.14 ± 0.28 | 2.17 ± 0.55 |
| 25 | Agulhas Current | WBC | 189 | 239 | -1.20 ± 0.09 | -0.58 | -2.730 ± 0.206 | -1.664 | -0.53 ± 0.05 | -1.21 ± 0.12 | 0.13 ± 0.01 |
| 26 | Tropical W Indian | Tropical | 46 | 68 | -0.06 ± 0.08 | 1.00 | -0.031 ± 0.044 | 0.815 | -0.16 ± 0.04 | -0.09 ± 0.03 | 0.20 ± 0.04 |
| 27 | W Arabian Sea | Indian Margins | 82 | 92 | 0.35 ± 0.04 | 1.14 | 0.342 ± 0.043 | 1.257 | -0.31 ± 0.04 | -0.31 ± 0.04 | 0.12 ± 0.04 |
| 28 | Red Sea | Marginal Sea | 158 | 174 | 0.24 ± 0.03 | 0.16 | 0.460 ± 0.065 | 0.330 | -0.15 ± 0.01 | -0.28 ± 0.02 | 0.57 ± 0.15 |
| 29 | Persian Gulf | Marginal Sea | 208 | 233 | 0.04 ± 0.08 | n.d. | 0.092 ± 0.203 | n.d. | -0.12 ± 0.02 | -0.31 ± 0.04 | 24.67 ± 12.09 |
| 30 | E Arabian Sea | Indian Margins | 298 | 317 | 0.21 ± 0.12 | 0.67 | 0.749 ± 0.427 | 2.555 | -0.30 ± 0.04 | -1.07 ± 0.15 | 0.67 ± 0.15 |
| 31 | Bay of Bengal | Indian Margins | 197 | 203 | -0.69 ± 0.12 | -0.22 | -1.641 ± 0.276 | -0.530 | -0.31 ± 0.04 | -0.74 ± 0.09 | 0.43 ± 0.11 |
| 32 | Tropical E Indian | Indian Margins | 727 | 763 | -0.06 ± 0.07 | -0.02 | -0.482 ± 0.569 | -0.170 | -0.20 ± 0.02 | -1.78 ± 0.17 | 0.50 ± 0.04 |
| 33 | Leeuwin Current | EBC | 81 | 117 | -2.05 ± 0.15 | -0.98 | -2.010 ± 0.148 | -1.379 | -0.60 ± 0.07 | -0.58 ± 0.07 | 0.56 ± 0.16 |
| 34 | S Australia | Subpolar | 392 | 436 | -1.37 ± 0.18 | -1.14 | -6.438 ± 0.859 | -5.983 | -0.27 ± 0.03 | -1.29 ± 0.14 | 0.74 ± 0.25 |
| 35 | E Australian Current | WBC | 98 | 130 | -1.74 ± 0.20 | -1.09 | -2.036 ± 0.205 | -1.695 | -0.50 ± 0.07 | -0.58 ± 0.08 | 0.37 ± 0.04 |
| 36 | New Zealand | Subpolar | 263 | 286 | -1.23 ± 0.16 | -1.25 | -3.882 ± 0.498 | -4.274 | -0.52 ± 0.07 | -1.64 ± 0.23 | 0.49 ± 0.04 |
| 37 | N Australia | Tropical | 2278 | 2292 | -0.29 ± 0.11 | 0.44 | -7.872 ± 3.114 | 12.120 | -0.23 ± 0.04 | -6.19 ± 1.00 | 0.38 ± 0.03 |
| 38 | S-E Asia | Tropical | 2130 | 2160 | -0.29 ± 0.07 | -0.91 | -7.344 ± 1.908 | -23.609 | -0.20 ± 0.03 | -5.01 ± 0.72 | 0.49 ± 0.05 |
| 39 | China Sea and Kuroshio | WBC | 1132 | 1129 | -1.99 ± 0.15 | -1.41 | -27.046 ± 1.991 | -19.100 | -0.45 ± 0.05 | -6.13 ± 0.72 | 0.32 ± 0.01 |
| 40 | Sea of Japan | Marginal Sea | 233 | 147 | -3.07 ± 0.17 | -3.47 | -8.613 ± 0.475 | -6.113 | -0.51 ± 0.06 | -1.44 ± 0.18 | 1.64 ± 0.24 |
| 41 | Sea of Okhotsk | Marginal Sea | 933 | 952 | -1.66 ± 0.07 | 1.31 | -18.623 ± 0.761 | 14.955 | -0.36 ± 0.03 | -4.00 ± 0.34 | 3.52 ± 1.38 |
| 42 | N-W Pacific | Subpolar | 1025 | 1000 | -1.85 ± 0.14 | -0.70 | -22.760 ± 1.726 | -8.419 | -0.24 ± 0.04 | -2.99 ± 0.52 | 1.48 ± 0.59 |
| 43 | Siberian Shelves | Polar | 1848 | 1889 | -0.47 ± 0.10 | -0.90 | -10.499 ± 2.117 | -20.322 | -0.05 ± 0.01 | -1.09 ± 0.28 | 4.10 ± 0.64 |
| 44 | Barents and Kara Seas | Polar | 1559 | 1680 | -0.75 ± 0.14 | -1.60 | -14.176 ± 2.585 | -32.225 | -0.11 ± 0.02 | -2.05 ± 0.43 | 1.58 ± 0.46 |
| 45 | Antarctic Shelves | Polar | 2452 | 2936 | -0.90 ± 0.14 | -0.15 | -26.630 ± 3.989 | -5.381 | -0.69 ± 0.07 | -20.30 ± 2.18 | 2.08 ± 0.29 |

**Table 3.** Weighted mean of simulated and data-based sea-to-air $CO_2$ fluxes and simulated residence time for each MARCATS class, excluding the Sea of Okhotsk (see text). Abbreviations are included for Eastern Boundary Current (EBC) and Western Boundary Current (WBC).

| Class | Sea-to-air $CO_2$ flux $(\mathrm{mol\,C\,m^{-2}\,yr^{-1}})$ | | | Residence |
|---|---|---|---|---|
| | Total (LA14) | Total (model) | Anthropogenic (model) | time (month) |
| EBC | 0.12 | -0.12 ± 0.16 | -0.42 ± 0.03 | 1.52 ± 0.22 |
| Indian margins | 0.19 | -0.06 ± 0.05 | -0.24 ± 0.02 | 0.49 ± 0.04 |
| Marginal Seas | -0.56 | -0.92 ± 0.07 | -0.29 ± 0.01 | 10.34 ± 3.50 |
| Polar Margins | -0.88 | -0.83 ± 0.06 | -0.32 ± 0.03 | 2.18 ± 0.20 |
| Subpolar Margins | -1.23 | -1.61 ± 0.07 | -0.36 ± 0.02 | 0.92 ± 0.16 |
| Tropical Margins | -0.10 | -0.15 ± 0.06 | -0.22 ± 0.02 | 0.42 ± 0.03 |
| WBC | -0.80 | -1.65 ± 0.08 | -0.48 ± 0.03 | 0.31 ± 0.01 |

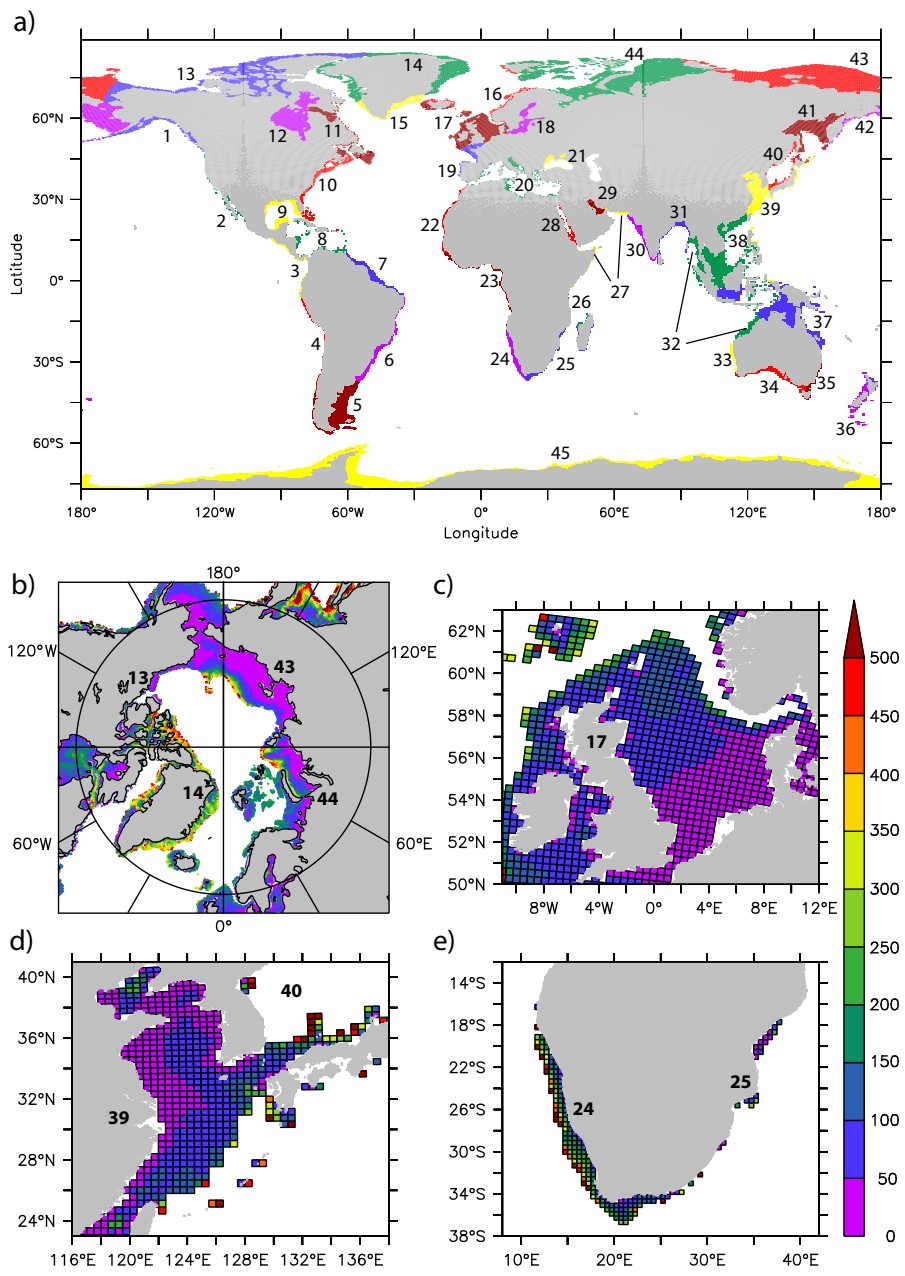

**Figure 1.** (a) Global segmentation of the coastal ocean following Laruelle et al. (2013) as regridded on the ORCA05 model grid. Colors distinguish limits between the MARCATS regions; numbers indicate regions defined in LA13. To perceive the spatial resolution of the ORCA05 configuration in the MARCATS context, we show zooms of bathymetry in 4 regions: (b) The Arctic polar margins, (c) the North Sea, (d) the Sea of Japan, the China Sea, and Kuroshio, and (e) Southern Western Africa and the Aghulas Current. In the latter 3 panels, grid resolution is indicated by thin black lines.

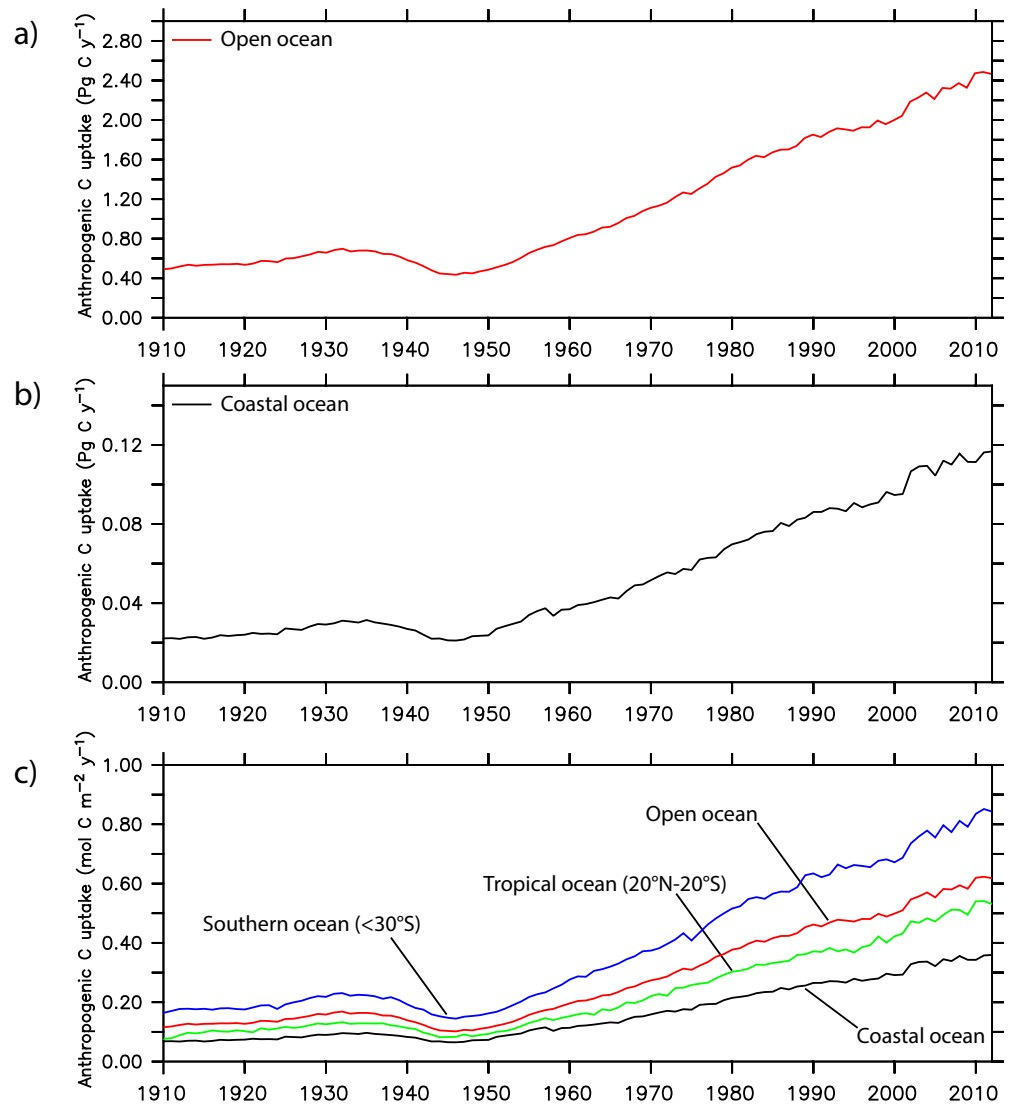

**Figure 2.** Simulated temporal evolution of area-integrated anthropogenic carbon uptake for (a) the open ocean and (b) the coastal ocean. (c) Analogous evolution of anthropogenic carbon uptake for the open ocean, the coastal ocean, the Southern Ocean, and the tropical oceans, but given as the average flux per unit area.

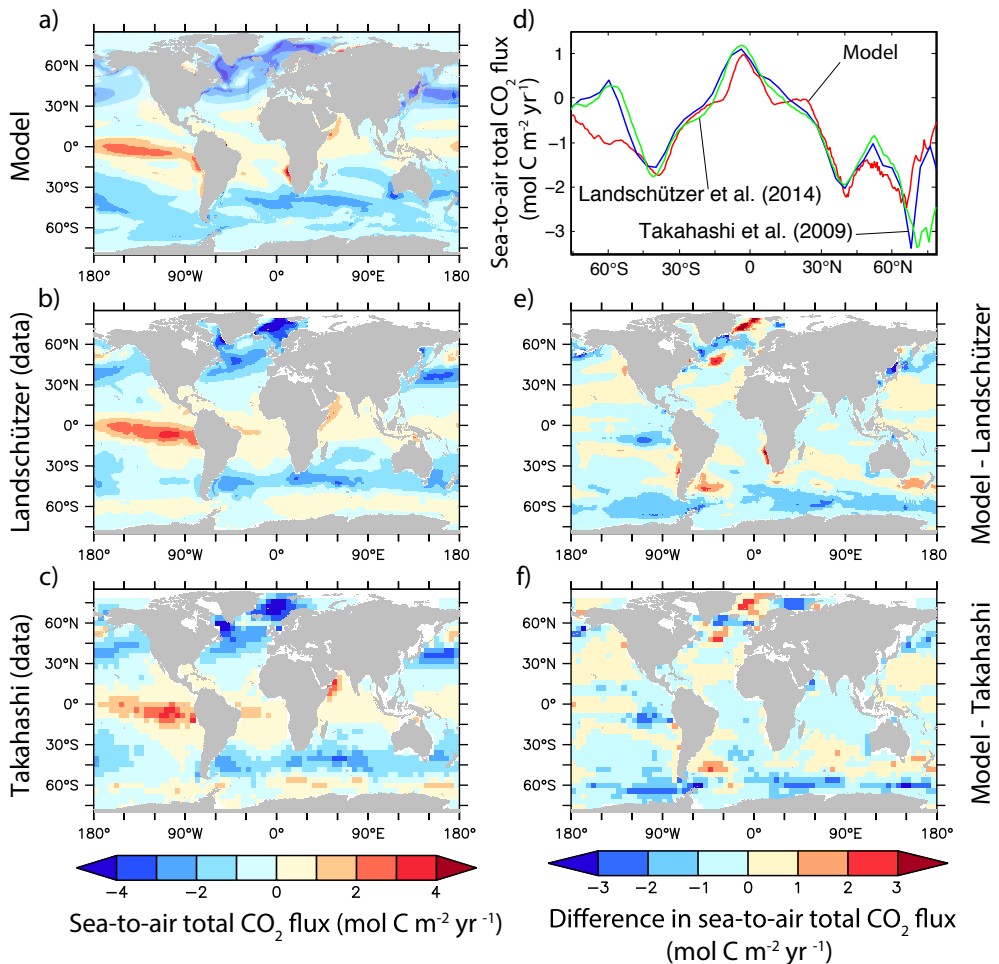

**Figure 3.** Climatological mean of sea-to-air flux of total carbon fluxes in $\mathrm{mol\,C\,m^{-2}\,y^{-1}}$ for (a) the model average during 1993–2012, (b) the data-based estimate from Landschützer et al. (2014) for 1998–2011, and (c) the data-based estimate from Takahashi et al. (2009) for the 2000–2009. Panels (d) and (f) present differences between simulated and observed sea-to-air total carbon fluxes $(\mathrm{mol\,C\,m^{-2}\,yr^{-1}})$ relative to (b) and (c), respectively. d) presents the latitudinal distribution of the simulated and the observed mean sea-to-air total carbon fluxes.

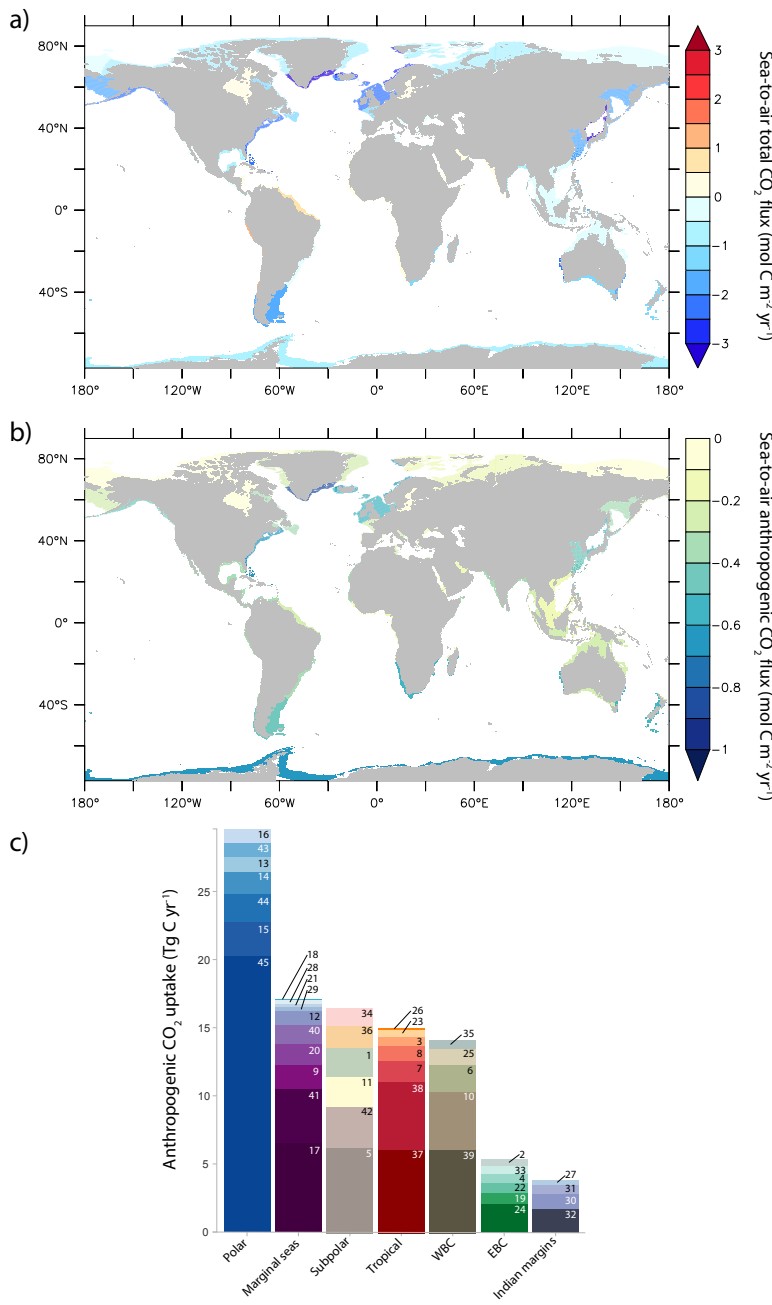

**Figure 4.** Global mean distribution of the simulated sea-to-air flux of (a) total carbon and (b) anthropogenic carbon over 1993–2012 as $\mathrm{mol\,C\,m^{-2}\,yr^{-1}}$ in the global coastal ocean segmented following MARCATS from LA13. (c) Bar chart of the anthropogenic carbon uptake in $\mathrm{Tg\,C\,yr^{-1}}$ according to the MARCATS classification. Abbreviations are included for Eastern Boundary Current (EBC) and Western Boundary Current (WBC). Links between numbers and regions are reported in Table 2. Interactive illustrations can be found at http://lsce-datavisgroup.github.io/CoastalCO2Flux/.

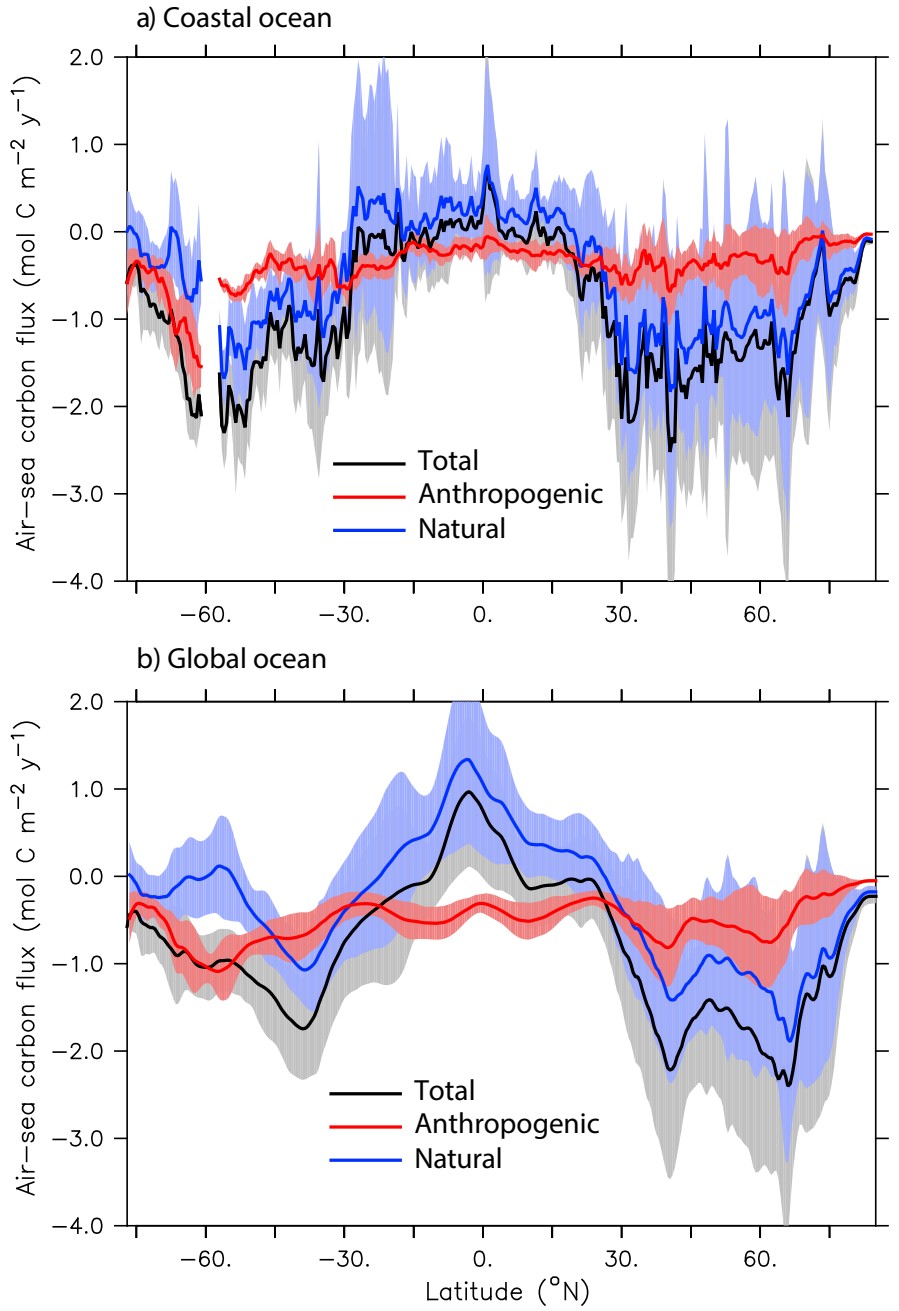

**Figure 5.** Zonal-mean, sea-to-air fluxes of total, anthropogenic, and natural $CO_2$ $(\mathrm{mol\,C\,m^{-2}\,yr^{-1}})$ given as the average over 1993–2012 for (a) the coastal ocean and (b) the global ocean. Shaded areas indicate the standard deviation of environmental variability of all ocean grid cells within each latitudinal band. Interannual variations are not shown.

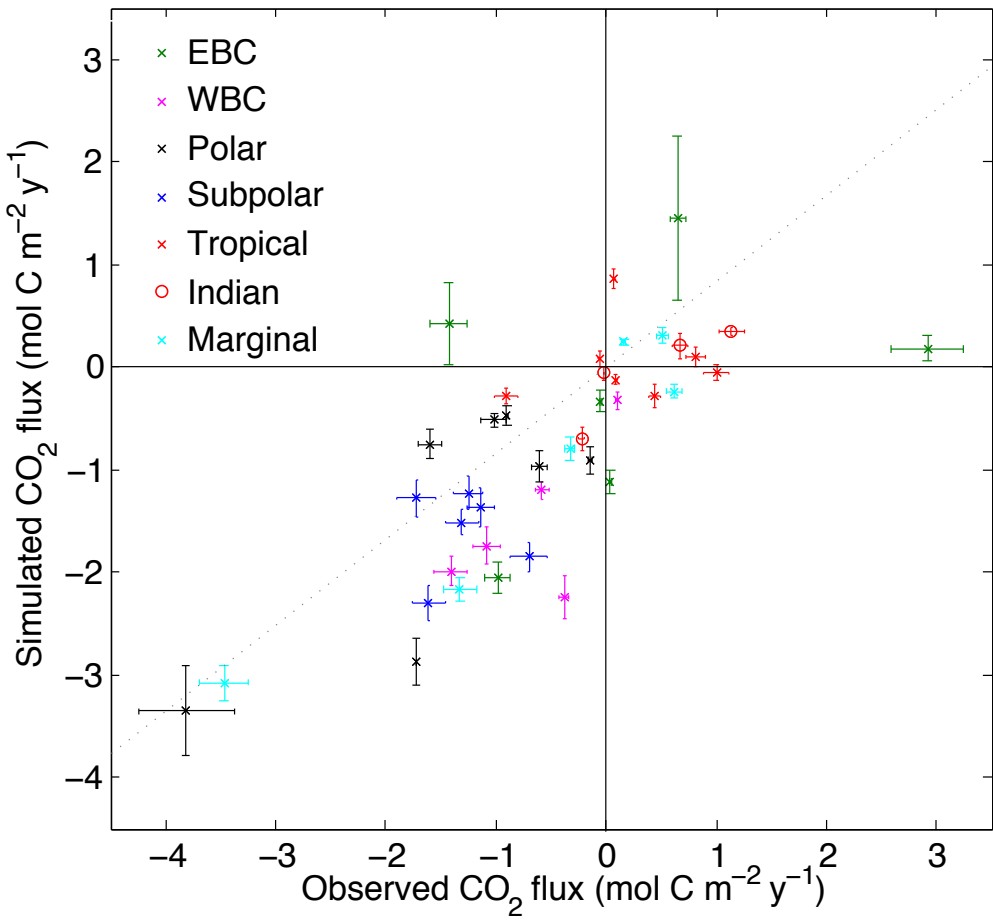

**Figure 6.** Simulated versus observed MARCATS sea-to-air flux of total carbon in (a) $\mathrm{mol\,C\,m^{-2}\,yr^{-1}}$ and (b) $\mathrm{Tg\,C\,yr^{-1}}$. Vertical error bars show the standard deviation from the 1993–2012 interannual variability for model results and the horizontal bars correspond to the 1990–2011 variability from computational methods used in LA14 for observation-based estimates. Here, regression line (grey dotted) have $y$-intercepts forced to 0. All MARCATS regions have been used except the Black Sea, the Persian Gulf (no data estimate), and the Sea of Okhotsk (see text).

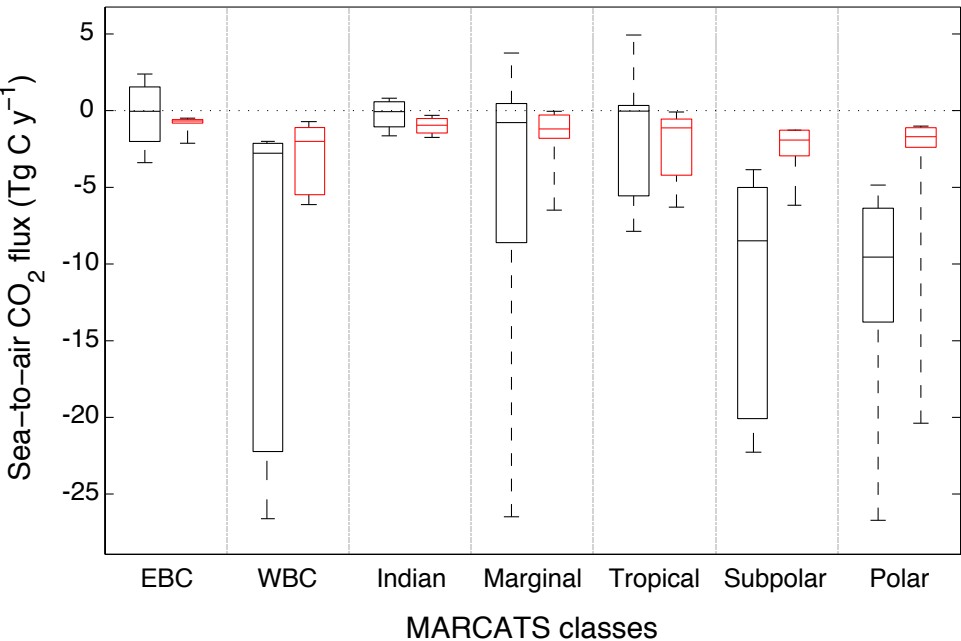

**Figure 7.** Box plots of the simulated sea-to-air $CO_2$ fluxes ($\mathrm{Tg\,C\,yr^{-1}}$) grouped into the MARCATS classes of the coastal ocean. Black boxes indicate total fluxes; red boxes indicate anthropogenic fluxes. Shown are the lowest estimate, the first quartile, the median, the third quartile, and the highest estimate for each class.

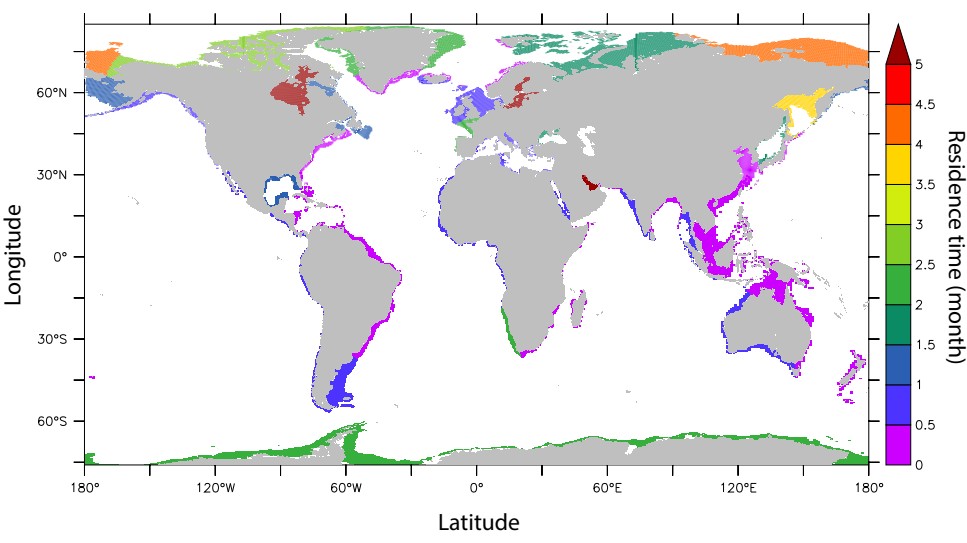

**Figure 8.** Global distribution of simulated residence time (month) for the global coastal ocean segmented following Laruelle et al. (2013).

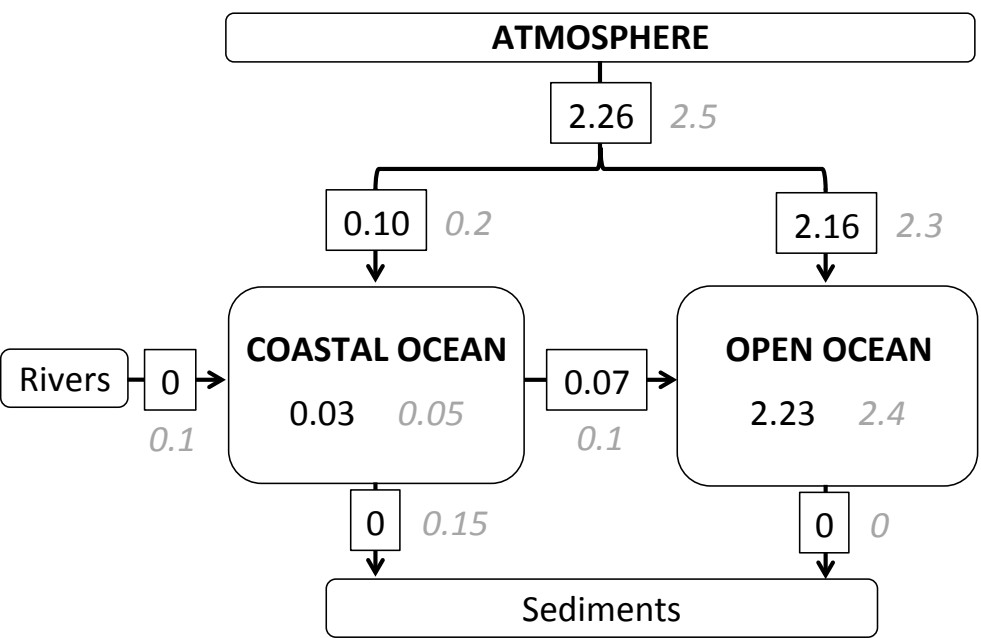

**Figure 9.** Transfer of anthropogenic carbon between the atmosphere, coastal ocean, and open ocean along with increases in the corresponding inventory in each reservoir, given as the average of simulated values over 1993–2012. All results are in $\mathrm{Pg\,C\,yr^{-1}}$. Simulated results are shown as dark numbers in boxes; adjacent numbers (grey italic) indicate data-based estimates for the 2000–2010 average (Regnier et al., 2013).

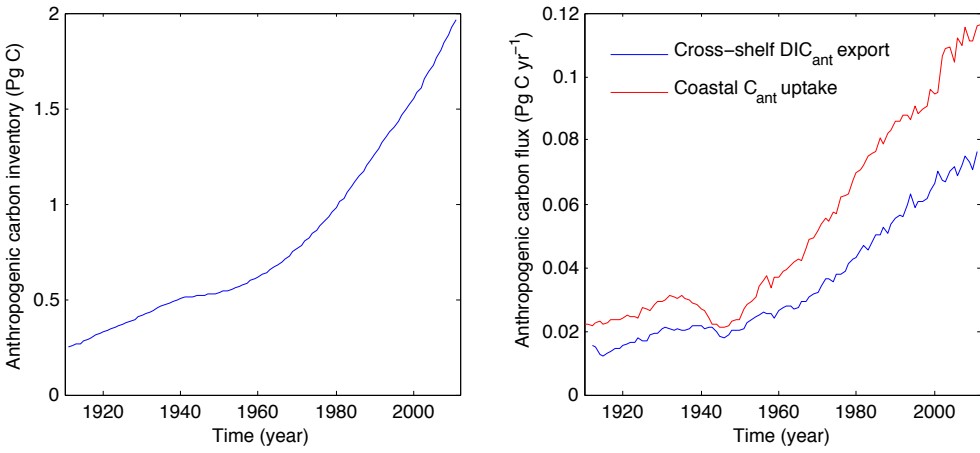

**Figure 10.** Simulated temporal evolution of (a) coastal-ocean inventory of anthropogenic carbon given in $\mathrm{Pg\,C}$ and (b) anthropogenic $CO_2$ ($C_{ant}$) uptake by the global coastal ocean and global cross-shelf export of anthropogenic carbon ($DIC_{ant}$) given in $\mathrm{Pg\,C\,yr^{-1}}$.