# Peer review of "Coastal-ocean uptake of anthropogenic carbon"

_Biogeosciences, 2016_

## Referee Comment (RC1) · P. Regnier (Referee) · 18 Apr 2016

The ms. by Bourgeois et al. is the very first attempt to quantify the air-sea $CO_2$ flux for the global coastal ocean using a highly-resolved 3D model. The authors compare in a convincing way their model results with observational data and discuss in detail the obtained spatial variability in the air-water $CO_2$ exchange. The approach is well described and model results are solid ; I am thus very supportive of this research. In addition, the authors have attempted a quantification of the anthropogenic perturbation on the air-sea $CO_2$ flux, with the key finding that the magnitude of the perturbation could be significantly smaller than previously taught. This is obviously an important result that further strengthen the value of this contribution. However, the latter aspect has several shortcomings that I believe need to be adressed fully (see in particular major comments 2 and 3) before publication.
Major comments

1) Uncertainties are only reported once for the anthropogenic CO2 flux (0.1 +- 0.01). You need to explain how this uncertainty has been estimated. It is also much lower than the uncertainty on the total simulated flux (0.27 +- 0.07), which is quite surprising. More generally, uncertainties and their quantification method should be reported for all fluxes and consistently throughout the text.

2) Section 4.1.2 on anthropogenic fluxes provides a suitable comparison with previous estimates. However, the last paragraph is misleading as one of the key reason why the size of the perturbation could be larger in Mackenzie and co-workers is the stimulation of the biological pump by enhanced land-derived nutrient inputs. These aspects should be included in the discussion, but also much earlier in the text (introduction and, eventually, title). That is, the authors should clearly state right from the start that they only consider atmopheric CO2 as their sole anthropogenic driver. As a result, I believe that only the physical dissolution pump is impacted, i.e., the model should simulate constant net ecosystem productivity (NEP) and (I suspect) constant net ecosystem calcification (NEC) during the entire historical period. The values for NEP and NEC should be reported and discussed (a subject of intense debate within the coastal C community) as this could be (another) plausible reason for the discrepancy with earlier estimates. Finally, nothing is said about temperature effects on the uptake of CO2. This aspect should also be included in the description/discussion.

3) Section 4.2 provides an explanation for the smaller relative magnitude of the global coastal anthropogenic CO2 uptake compared to the global ocean. As it is, Figs. 2, 5 and 8 do not satisfactorily substantiate the proposed mechanism. What is missing are plots of temporal evolution of (organic and inorganic) carbon accumulation (also % relative increase) and cross-shelf export for the entire simulation period. The Revelle factors shoud also be reported. Based on the proposed mechanism, I would suspect to see a progressive decrease of the ratio of anthropogenic carbon uptake of the coastal ocean to the global carbon uptake due to the accumulation of anthopogenic CO2 in

the coastal water column through time and this does not seem to be the case (Fig.2). I also would suspect to see a progressive increase in the Revelle factor (faster for the coastal ocean than open ocean) through time. In addition, it would be interesting to briefly discuss why the uptake fluxes per unit surface area for the shelf seem to be larger than for the open ocean under pre-industrial conditions (fig.5). Furthermore, the authors should report the calculated horizontal cross-shelf transport of water as this is a crucial number to sustain their conclusion (a first sensitivity analysis could have been useful in this context). Finally, I believe that comment 2 above (focus on the physical dissolution pump only) is also relevant in the context of section 4.2 -

Other comments

Abstract and conclusion : the authors should also summarize the main results on the total fluxes as this is the first time that a model-data comparion is performed with a physically-resolved model at the global scale.

Abstract page 1, line 8 : a high resolution is required not only to resolve the bathymetry, but also the complex coastal currents (which in my opinion are not all induced by the bathymetry)

Page 2 line 4 : I suspect that the word export refers to 'export production'. I would clarify because in the context of this paper, it could also refer to 'cross-shelf export'

Page 2 line 5 : the carbon export and burial fluxes are highly uncertain – see, e.g. Krumins et al. 2013 (BG) for a review. The same also holds for the productivity (even the sign of the NEP is uncertain – see Bauer et al., 2013. It is thus not correct to state that the air-sea $CO_2$ flux is the most uncertain of the C fluxes for the coastal ocean.

Page 2 line 15-16 : I agree about the $CO_2$ switch of the coastal ocean from source to sink, but do not agree fully with the proposed attribution. MacKenzie and co-workers highlight the change in NEP (from enhanced land nutrient inputs) as one of their key driver to explain the shift (see aslo Regnier et al., 2013 & Bauer et al., 2013 – for

reviews). Please clarify (see also major comments).

Page 3, line 20 : I recommend making reference to the few published long time series of $CO_2$ observations (> 1 decade) for the coastal ocean. I agree nevertheless that these time series alone are sparse and short. Thus, an observation-based global extrapolation of the anthropogenic component is highly uncertain

Page 4 line 24 : calcite particles are included. This is not a satisfactory description. Please state clearly if your model accounts for calcification as this process has a potentially important impact on the air-sea $CO_2$ exchange (see also major comments).

Page 4 line 29 : do you mean atmospheric deposition ?

Page 6 line 4 : Assuming that land derived DOC is entirely labile is a strong assumption. The flux (0.15 PgC yr-1 from the top of my head) is also significant. Thus, the extend to which your results depend on this assumption has to be discussed.

Page 6 lines 11-19 : The implication of a model outside of ' equilibrium ' has to be addressed. For instance, when you refer to a global ocean anthropogenic uptake of 2.3 PgC/yr-1, this number is obtained with a natural flux of -0,33 PgC yr-1 for the natural flux. Correct ?

Page 7, section 2.4 evaluation dataset : To leave no ambiguity, did you compare your model results with LA14 using the Wanninkhoff 1992 formulation or the updated formulation ?

Page 9, lines 12-14 : I would say 'weak carbon sources' and 'strong carbon sinks'

Page 9 lines 15-25: The phrasing is misleading (' our model results tend to underestimate total carbon flux, with 76% of the simulated specific fluxes lower than the data-based estimates '), as the absolute fluxes are actually larger in the model (i.e. larger negative sinks). ' Likewise ' is also not appropriate because the Arctic region is in fact the only latitudinal band where the model results predict a smaller sink than the observations. More broadly, I find that the results are quite comparable for the southern

hemisphere and the low latitude regions, but that discrepancies are significantly larger in the Norther hemisphere with a stronger sink modeled for the 30-60°N and a weaker sink modeled for the > 60°N latidudinal band (see also Fig 3 of LA14). Also, the fact that the areal-integrated fluxes show a weaker obs-model correlation than the fluxes per unit surface area requires discussion.

Page 10 line 14 : what do you mean by ' top two regions ' ?

Page 10 line 25 and further: It is important to state that (to my knowledge) only LA14 accounts for the sea-ice cover in the global estimates - this is an important effect on the quantification.

Page 11 line 5-10 : I agree that the exclusion of the proximal zone in the model assessment should have an impact on the sign of the flux under pre-industrial conditions. But what about the effect of the initialisation (the value of the sink is not reported for the coastal ocean in 1850) ? Stated differently, is the global coastal ocean in equilibrium at the onset of the simulations ? Regarding the proximal zone, bays, estuaries, deltas, lagoons are indeed sources of CO2 (see Laruelle et al., 2013 for the latest synthesis), banks should be too (a reference would be useful), but I am not sure about what is meant by ' marine wetlands '. If this refers to marshes and mangroves, they are then believe to be sinks for atm CO2 (see Cai, 2011, Regnier et al., 2013, Bauer et al., 2013). Thus, clarification is required here.

Page 11, section 4.1.2 first paragraph : I believe that regional scale studies have attempted an estimation of the anthropogenic CO2 uptake in EBUS. If true, they should be included in the discussion.

Page 11 line 21-22 : This sentence has to be rephrased as it implies that one modeling approach performs better than another. Please tone down.

Page 12, section 4.2 : the first two paragraphs on total fluxes should be merged with section 4.1.1 – Regarding the Amazon, what is the potential impact of assuming that

all the DOC (a large flux) is transformed into DIC in this region ? More generally, do you assume that this instantaneous transformation has no impact on alkalinity ?

Page 12, section 4.2 : the latidudinal trends in anthropogenic CO2 fluxes are also very similar for the coastal and open ocean (Figure 5). This aspect needs to be discussed.

Page 13, lines 12-19 : The computation of Revelle factor values is interesting, but it is important to stress that (to my knowledge), a higher value for the global coastal ocean compared to the global ocean remains highly speculative as this has not been demonstrated from observational data. Also, - and this is an important point – the sentence ' That finding is consistent with the lower simulated specific fluxes into the coastal ocean ' is not convincing. At the end, the Revelle factor should influence the total fluxes (and not its anthropogenic component) for which the area-based estimates indicate significantly larger negative sinks than in the global ocean (fig.5), i.e. the opposite of the anthropogenic component fluxes.

Page 13 lines 23-24 : The chemical factors are presented as independent of the physical factors controlling the air-sea CO2 exchange. However, based on the model construct, I feel that the higher Revelle factor for the coastal ocean precisely results from the physics of the coastal zone, with a progressive accumulation of DIC due to weaker cross-shelf export than CO2 air-sea exchange.

Figure 8 : I assume that fluxes refer to total anthropogenic fluxes, i.e. organic plus inorganic carbon – please clarify.

Spelling

Page 1, Line 10-11 rephrase – this sentence is odd

Page 2 line 4 : remove 2nd 'relative'

Page 4 line 25 : not sure that 'model' can be used as a verb

Page 10 line 19 : remove 'that'

Page 14 line 14 : remove 'of'

Page 14 line 24-5 : parenthesis wrongly placed

---

## Referee Comment (RC2) · Prof GRUBER (Referee) · 26 Apr 2016

**1  Summary**

Bourgeois et al. use an eddy-permitting global ocean biogeochemistry model to investigate the relative contribution of the ocean margins to the uptake of anthropogenic $CO_2$ from the atmosphere. They find that these regions take up a disproportionally low amount of anthropogenic $CO_2$, i.e., only 4.5% relative to this regions areal contribution of 7.5%. The authors suggest that it is the limited degree to which the anthropogenic $CO_2$ is transported and mixed offshore that leads to this low uptake. These results are in stark contrast to earlier studies based primarily on a few point observations that suggested a very high coastal ocean uptake of $CO_2$ from the atmosphere.

**2 Evaluation**

Understanding and predicting the future evolution of the oceanic sink for atmospheric $CO_2$ is of paramount importance for determining how much $CO_2$ we can emit in the coming decades without exceeding any climate target. While our confidence in the net exchange fluxes of $CO_2$ over the open ocean has increased substantially in recent years thanks to better observations and novel methods to interpret these data, our ability to constrain the fluxes along the continental margins has not increased commensurably. The existing observational constraints are still relatively weak and associated with sizeable uncertainties. By far the most extensive and detailed assessment to date by Laruelle et al. suggested a relatively small global net uptake of atmospheric $CO_2$, but their data-based approach did not permit them to separate this net flux into its "natural" and "anthropogenic" components.

Thus the study by Bourgeois et al. is much welcomed as it brings a consistent global perspective to the problem. Their model-based approach is far from perfect, owing to many issues ranging from its still coarse resolution to the limited of consideration of several processes that are of relevance in coastal systems (e.g., limited consideration of benthic processes), but by using - for the first time - an eddy permitting 3D model, it is a big step up from previous model based approaches that used highly simplified models. I particularly like that the authors spend a considerable amount of effort in order to assess their model's performance against observations and that they run two parallel simulations in order to determine the anthropogenic $CO_2$ uptake explicitly. The paper is overall well written, adequately illustrated and referenced. The discussion is generally thorough and the conclusions supported by the provided evidence including an appropriate consideration of the caveats. All in all, this is a very good study that is well suited for publication in *Biogeosciences*. However, before giving the green light, I would like the authors to consider my concerns regarding important model shortcomings.

- (i) *Model drift*: The model's drift is quite substantial, and I am not entirely convinced that the authors have fully considered the implications when discussing their results. I think the drift is large enough that it cannot be ignored.

- (ii) *: Reflective bottom boundary:* The model's lower boundary is assumed to "reflect" any settling organic matter back into the water column in remineralized form. Thus, this model does not incorporate any delayed response of remineralization nor any other benthic remineralization process of relevance such as benthic denitrification etc. This is a rather important omission, as these processes influence coastal biogeochemistry in many shallow marginal seas. I thus recommend to address and discuss this issue in somewhat more detail and to give it better consideration.

- (iii) *: Coastal-open ocean exchange:* This issue is actually addressed a bit more than the other two in the current version, but I still consider it worthwhile to assess the implications of this shortcoming in more detail.

**3 Recommendation**

I recommend acceptance of this manuscript with minor revisions.

**4 Minor comments**

Abstract, line 10: "absorbed by the coastal ocean": I suggest to define also in the abstract how the authors define the "coastal" ocean.

p1, line 15: "the ocean naturally mitigates..." I am not sure why the authors use the expression "naturally" here. I suggest to delete it.

p5, line 5, equation 2: The authors still use a coefficient of 0.30, while there are numerous studies that have shown that this coefficient needs to be lowered in order to close the oceanic C14 budget. It's time to change, no?

p5, lines 17 to 25: By neglecting the oceanic uptake of anthropogenic CO over the period 1750 until 1870, the authors underestimate the total anthropogenic uptake by about 10% or so. This should be taken into consideration more explicitly. I also suggest to avoid the use of the term "preindustrial" when referring to 1870.

p6, line 14: "as compared to the estimate of natural carbon outgassing of 0.45 Pg C yr-1". I find this comparison confusing, since the two processes are clearly distinct. The first number is the drift of the model, while the second one refers to the outgassing of river-derived carbon from the ocean. I do not think that these two numbers can be compared as done here.

p8, line 14: "Regionally, [the] overall patterns in the air-sea $CO_2$ flux are similar between the ..." How was the large model drift considered when making this statement? This issue is mentioned further down (line 24), but not really elaborated.

p9, lines 16-17: "Correlation [..] only 0.5 for area-integrated fluxes". How important is the remapping error here? (see my comment on Table 2).

p11, lines 23-24 ".. carbon sink in the Amazon river plume [..] is not reproduced". I think that this also has something to do with the model's shortcoming with regard to the benthic-water column interactions.

p14, lines 1-3: "... is transported offshore to the deeper open ocean". This deserves a little more discussion, particularly since a few lines below (e.g., lines 16-17) this issue is identified as "a critical question".

Table 2: some of the areal discrepancies are huge. I thus think that one needs to be very careful when comparing areally integrated fluxes. In fact, this should not be done, in my opinion, if the areas differ by more than let's say 10%.
Figure 2: reduction in uptake during the 1940s. Where is this coming from? To me, this looks like the model is overly sensitive to the rate of change in atm. CO2, perhaps due to the drift.

Figure 6: See also my comment about Table 2. I am not sure whether it is really appropriate to compare the areally integrated fluxes when the areas are that different to begin with.

Nicolas Gruber, April 2016

---

## Author Comment (AC1) · 31 May 2016

We thank Prof. Pierre Regnier (referee #1) and Prof. Nicolas Gruber (referee #2) for their comments and suggestions.

We first comment on 3 key topics raised by the referees, and then provide point-by-point responses for each referee. The 3 key topics are (1) the need for more information to support our proposed mechanism for lower coastal anthropogenic $CO_2$ uptake due to limitation by cross-shelf exchange, (2) the need to clarify our estimate of anthropogenic carbon fluxes and on the role of potential biological changes, and (3) the effect of model drift on our results.

**1) Cross-shelf exchange**

We concur with both referees that more information is needed to support our conclusion that it is inadequate cross-shelf exchange that reduces the coastal ocean's uptake of anthropogenic CO2 (per unit area) relative to that of the global ocean.

Following the suggestion by Referee #1, we plan to include a new figure with time series of $C_{ant}$ uptake, $C_{ant}$ storage, and cross-shelf export of $C_{ant}$ (Figure 1 in this response). This new figure demonstrates clearly that simulated cross-shelf export of dissolved inorganic anthropogenic carbon ($DIC_{ant}$) is less than the simulated anthropogenic $CO_2$ uptake for the coastal ocean, implying an accumulation of $DIC_{ant}$ in the coastal waters column during the simulation.

As suggested by the Referee #2, we will also offer more detail about the simulated coastal-ocean ocean exchange. For that, we will add the simulated water residence time for each MARCATS region to Tables 2 and 3 of the revised manuscript and we will introduce a new figure (Figure 2 in this response). These new results reveal that simulated residence times for most coastal regions are of the order of a few months or less, except for Hudson Bay, the Baltic Sea and the Persian Gulf. The latter three regions are generally more confined and we would expect longer residence times, although our model simulations were never designed to simulate these regions accurately. Generally then, our simulated residence times are shorter than what has been published for similarly defined coastal regions (see Jickells et al. (1998), Men et al. (2015)). The absence of tides and the spatial resolution used in our study do not permit to reproduce the mesoscale variability of the coastal circulation (e.g. eddies and upwellings). Depending on coastal regions, these processes may increase/decrease residence time. The comparison with these published estimates is also difficult since we average residence time estimates on extended areas.

Jickells et al. (1998)
http://science.sciencemag.org/content/281/5374/217
Men et al. (2015)
http://link.springer.com/article/10.1007%2Fs10967-014-3749-y

[Figure]

*Figure 1: Simulated temporal evolution of (a) cross-shelf export for anthropogenic dissolved inorganic carbon (DIC$_{ant}$) in blue and area-integrated anthropogenic CO$_2$ uptake for the coastal ocean in red, given in Pg C yr$^{-1}$, and (b) DIC$_{ant}$ inventory for the coastal ocean as Pg C. Missing years will be added in revised manuscript.*

[Figure]

*Figure 2: Global distribution of simulated residence time (month) for the global coastal ocean segmented following Laruelle et al. (2013). Results are also reported on updated Tables 2 and 3 at the end of this document.*

All these points will be added to the revised manuscript.

**2) Our estimate of anthropogenic carbon fluxes and the role of potential biological changes**

Our method to compute anthropogenic air-sea CO$_2$ fluxes (FCO$_2$) is based on the difference between total FCO$_2$ from an historical simulation (with increasing atmospheric CO$_2$) and natural FCO$_2$ from a control simulation (i.e., with the same physical forcing but with fixed pre-industrial atmospheric CO$_2$). This method is commonly used by the modeling community to estimate anthropogenic carbon fluxes (e.g. http://ocmip5.ipsl.jussieu.fr/OCMIP/ or Bopp et al., 2015). By definition, our anthropogenic air-sea CO$_2$ fluxes only respond to increasing

atmospheric $CO_2$. They do not include any effect from potential changes in ocean physics or biology, because those changes are identical in the historical and the control simulations. Hence, even if surface temperature or biological fluxes (NEP or NEC) change in response to the forcing, they do not impact anthropogenic carbon uptake per se.

That said, we agree with the reviewers that potential changes in the physics and biology as well as changes in riverine input or in the interactions with the sediment may be of primary importance, would modify the distribution of total carbon and alkalinity, and hence would also modify the potential of the coastal ocean to absorb anthropogenic carbon.

These points will be discussed in more detail in the revised manuscript.

Bopp et al. (2015)
http://dx.doi.org/10.1002/2015GL065073

**3) Model drift and potential implications.**

We concur with both reviewers that details about model drift are important. In the original manuscript we stated the following: "At lower resolution (ORCA2), after a spin up of 3000 years, there is 0.26 Pg C yr-1 greater globally integrated sea-to-air flux, relative to results after only a 50-year spin up. Nearly all of that enhanced sea-to-air CO2 flux due to the longer spin up comes from the Southern Ocean." Unfortunately, such spin-up length is currently out of reach regarding to computation costs of our ORCA05-PISCES configuration. We emphasize though that our anthropogenic $FCO_2$ estimates are expected to be influenced very little by model drift because of the way anthropogenic carbon is defined ($C_{ant} = C_{total} - C_{natural}$), i.e., drift affects both $C_{total}$ and $C_{natural}$ in the same way.

**POINT-BY-POINT REPLY**

- Referee #1 comments

The manuscript by Bourgeois et al. is the very first attempt to quantify the air-sea $CO_2$ flux for the global coastal ocean using a highly-resolved 3D model. The authors compare in a convincing way their model results with observational data and discuss in detail the obtained spatial variability in the air-water $CO_2$ exchange. The approach is well described and model results are solid; I am thus very supportive of this research. In addition, the authors have attempted a quantification of the anthropogenic perturbation on the air-sea $CO_2$ flux, with the key finding that the magnitude of the perturbation could be significantly smaller than previously taught. This is obviously an important result that further strengthens the value of this contribution. However, the latter aspect has several shortcomings that I believe need to be addressed fully (see in particular major comments 2) and 3) before publication.

Thanks for this positive general feedback.

Major comments

1) Uncertainties are only reported once for the anthropogenic $CO_2$ flux (0.1 +- 0.01). You need to explain how this uncertainty has been estimated. It is also much lower than the uncertainty on the total simulated flux (0.27 +- 0.07), which is quite surprising. More

generally, uncertainties and their quantification method should be reported for all fluxes and consistently throughout the text.

The uncertainty on our estimate of the coastal-ocean uptake of anthropogenic carbon is defined as its interannual standard deviation over 1993-2012. This will be clarified to the revised manuscript; uncertainties will also be given for all fluxes (see updated Tables 2 and 3 at the end of our response).

Since our uncertainty values are based on the interannual variability over 1993-2012, the uncertainty on the simulated total $FCO_2$ is much higher than that of the anthropogenic flux. That is, the strong variability of total $CO_2$ flux is due almost entirely to variability in its natural component so that the variability in their difference (anthropogenic flux) is small.

Another way to estimate uncertainties for such fluxes would be to use multiple models or sensitivity tests with one model where key parameters were varied. These exercises are currently out of reach and left for future work.

2) Section 4.1.2 on anthropogenic fluxes provides a suitable comparison with previous estimates. However, the last paragraph is misleading as one of the key reason why the size of the perturbation could be larger in Mackenzie and co-workers is the stimulation of the biological pump by enhanced land-derived nutrient inputs. These aspects should be included in the discussion, but also much earlier in the text (introduction and, eventually, title). That is, the authors should clearly state right from the start that they only consider atmospheric $CO_2$ as their sole anthropogenic driver. As a result, I believe that only the physical dissolution pump is impacted, i.e., the model should simulate constant net ecosystem productivity (NEP) and (I suspect) constant net ecosystem calcification (NEC) during the entire historical period. The values for NEP and NEC should be reported and discussed (a subject of intense debate within the coastal C community) as this could be (another) plausible reason for the discrepancy with earlier estimates. Finally, nothing is said about temperature effects on the uptake of $CO_2$. This aspect should also be included in the description/discussion.

We agree with the reviewer that the way we define the anthropogenic flux hampers any strict comparison with some previous estimates in which other terms are taken into account. This will be clarified in the revised manuscript, and we refer the reviewer here only to our general reply above for a first response to this comment.

3) Section 4.2 provides an explanation for the smaller relative magnitude of the global coastal anthropogenic $CO_2$ uptake compared to the global ocean. As it is, Figs. 2, 5 and 8 do not satisfactorily substantiate the proposed mechanism. What are missing are plots of temporal evolution of (organic and inorganic) carbon accumulation (also % relative increase) and cross-shelf export for the entire simulation period. The Revelle factors should also be reported. Based on the proposed mechanism, I would suspect to see a progressive decrease of the ratio of anthropogenic carbon uptake of the coastal ocean to the global carbon uptake due to the accumulation of anthropogenic $CO_2$ in the coastal water column through time and this does not seem to be the case (Fig.2). I also would suspect to see a progressive increase in the Revelle factor (faster for the coastal ocean than open ocean) through time. In addition, it would be interesting to briefly discuss why the uptake fluxes per unit surface area for the shelf seem to be larger than for the open ocean under pre-industrial conditions (Fig. 5). Furthermore, the authors should report the calculated horizontal cross-shelf transport of water as this is a crucial number to sustain their conclusion (a first sensitivity analysis could have

been useful in this context). Finally, I believe that comment 2 above (focus on the physical dissolution pump only) is also relevant in the context of section 4.2

We concur with the reviewer that more information is indeed needed to substantiate our proposed mechanism; we refer to the general reply (point 1, above) in which we indicate the additional figures and discussions that we will introduce in the revised manuscript.

We have also calculated the evolution of the Revelle factor for the coastal and open ocean over the historical period. While the coastal Revelle factor is 16% more than in the open ocean in 1870, that ratio evolves to being only 17% more for the last simulated decades (around 2000). Thus different rates of change of Revelle factor between the coastal and open ocean cannot solely explain the amplified anthropogenic carbon accumulation in the water column of the coastal ocean.

Finally, concerning the horizontal cross-shelf transport of water, we do not fully understand the suggestion of the referee. We have chosen to compute water residence times for each MARCATS. We refer the reviewers to our general reply for this comment.

Other comments

Abstract and conclusion: the authors should also summarize the main results on the total fluxes as this is the first time that a model-data comparison is performed with a physically-resolved model at the global scale.

As proposed, the main results on the model-data comparison of total fluxes will be added to the abstract and conclusion of the revised manuscript.

Abstract page 1, line 8: a high resolution is required not only to resolve the bathymetry, but also the complex coastal currents (which in my opinion are not all induced by the bathymetry)

"To begin to better resolve coastal bathymetry" will be replaced by "to begin to better resolve coastal bathymetry and take into account the complex coastal currents" in the revised manuscript.

Page 2 line 4: I suspect that the word export refers to 'export production'. I would clarify because in the context of this paper, it could also refer to 'cross-shelf export'

As the intended meaning was "cross-shelf export", "carbon export" will be replaced by "cross-shelf carbon export" to the revised manuscript to clarify this sentence.

Page 2 line 5: the carbon export and burial fluxes are highly uncertain – see, e.g. Krumins et al. 2013 (BG) for a review. The same also holds for the productivity (even the sign of the NEP is uncertain – see Bauer et al., 2013. It is thus not correct to state that the air-sea $CO_2$ flux is the most uncertain of the C fluxes for the coastal ocean.

We presume that the referee refers to page 2, line 9 rather than line 5. We agree that the "less is known" formulation for coastal-ocean air-sea $CO_2$ exchange is not suitable here. In the revised manuscript, the last sentence of this paragraph will be replaced by "All these

estimates suffer from high uncertainties as do those for coastal-ocean air-sea $CO_2$ exchange (Laruelle et al., 2014), particularly its anthropogenic component".

Page 2 line 15-16: I agree about the $CO_2$ switch of the coastal ocean from source to sink, but do not agree fully with the proposed attribution. Mackenzie and co-workers highlight the change in NEP (from enhanced land nutrient inputs) as one of their key driver to explain the shift (see also Regnier et al., 2013 & Bauer et al., 2013 – for reviews). Please clarify (see also major comments).

Referee #1 does well to emphasize the need to introduce the proposed role of the changing NEP and riverine nutrient inputs in the $CO_2$ source-to-sink shift of the coastal ocean from Mackenzie et al. (2004). This proposed mechanism will be discussed in the revised manuscript as will be that from Bauer et al. (2013) to explain the $CO_2$ source-to-sink shift due only to the increased physical uptake of atmospheric $CO_2$ (with constant NEP).

Page 3, line 20: I recommend making reference to the few published long time series of $CO_2$ observations (> 1 decade) for the coastal ocean. I agree nevertheless that these time series alone are sparse and short. Thus, an observation-based global extrapolation of the anthropogenic component is highly uncertain.

Two references dealing with long time series of $CO_2$ observations in the coastal ocean will be added in the revised manuscript (Astor et al., 2013 with 1996-2008 $CO_2$ data at CARIACO station on Venezuelan coasts and Ishii et al., 2011 with the 1994-2008 $CO_2$ data along 137°E on Japanese coasts) to support the lack of data-based estimates. To our knowledge, these are the only available time series of $CO_2$ data of more than a decade.

Astor et al. (2013)
http://dx.doi.org/10.1016/j.dsr2.2013.01.002
Ishii et al. (2011)
http://dx.doi.org/10.1029/2010JC006831

Page 4 line 24: calcite particles are included. This is not a satisfactory description. Please state clearly if your model accounts for calcification as this process has a potentially important impact on the air-sea $CO_2$ exchange (see also major comments).

We have chosen not to provide great detail on the model's calcification-related processes, which are described extensively elsewhere (Aumont and Bopp, 2006). The simulated NEC remains constant at a first order and does not respond to increasing anthropogenic $CO_2$. Moreover, calcification is identical in our historical and control simulations; hence, it does not affect the simulated anthropogenic carbon perturbation.

Page 4 line 29: do you mean atmospheric deposition?

Yes, "budgets" will be replaced by "deposition" in the revised manuscript.

Page 6 line 4: Assuming that land derived DOC is entirely labile is a strong assumption. The flux (0.15 PgC yr-1 from the top of my head) is also significant. Thus, the extent to which your results depend on this assumption has to be discussed.

Referee #1 is absolutely right that the representation of riverine DOC input in the model is extremely simple. Dissolved organic matter is indeed assumed to remineralize instantaneously at the river mouths, thus contributing to the DIN, DIP, DIC pools. Yet complexifying the approach will not affect our estimate of anthropogenic carbon uptake since the land-derived carbon delivery and its lability remains constant throughout simulation. We agree with the referee on the need to clarify this point in the revised manuscript.

Page 6 lines 11-19: The implication of a model outside of ' equilibrium ' has to be addressed. For instance, when you refer to a global ocean anthropogenic uptake of 2.3 PgC/yr-1, this number is obtained with a natural flux of -0,33 PgC yr-1 for the natural flux. Correct?

This is correct. We refer the reviewers to the general reply for this comment.

Page 7, section 2.4 evaluation dataset: To leave no ambiguity, did you compare your model results with LA14 using the Wanninkhoff 1992 formulation or the updated formulation?

We used Laruelle et al. (2014) estimates computed using the formulation of Wanninkhof (1992) as modified by Takahashi et al. (2009) to compare with simulated $CO_2$ fluxes computed using the initial Wanninkhof et al. (1992) formulation. Although the gas exchange formulation is critical for air-sea flux estimates derived from observations, it has little impact on model-derived fluxes of anthropogenic carbon (Sarmiento et al., 2002).

Sarmiento et al. (2002)
http://dx.doi.org/10.1063/1.1510279

Page 9, lines 12-14: I would say 'weak carbon sources' and 'strong carbon sinks'

We will make both of these changes in the revised manuscript.

Page 9 lines 15-25: The phrasing is misleading ('our model results tend to underestimate total carbon flux, with 76% of the simulated specific fluxes lower than the data-based estimates'), as the absolute fluxes are actually larger in the model (i.e. larger negative sinks). 'Likewise' is also not appropriate because the Arctic region is in fact the only latitudinal band where the model results predict a smaller sink than the observations. More broadly, I find that the results are quite comparable for the southern hemisphere and the low latitude regions, but that discrepancies are significantly larger in the Northern hemisphere with a stronger sink modeled for the 30-60∘ N and a weaker sink modeled for the > 60∘N latitudinal band (see also Fig 3 of LA14). Also, the fact that the areal-integrated fluxes show a weaker obs-model correlation than the fluxes per unit surface area requires discussion.

The sentence "our model results tend to underestimate total carbon flux, with 76% of the simulated specific fluxes lower than the data-based estimates" has been replaced by "our model results tend to simulate larger sinks / smaller sources than observed (i.e. 76% of the specific simulated fluxes of total carbon with lower relative values than the data-based estimates)".

"Likewise" will be replaced by "Otherwise".

Using the latitudinal distribution of natural fluxes, we can explain why carbon uptake is indeed larger in the coastal ocean. This is because upwelling zones (natural $CO_2$ sources) are

extremely restricted (i.e. narrow continental shelf) contrary to the $CO_2$ sink regions with commonly large continental shelves.

Concerning the weak obs-model correlation for area-integrated fluxes, remapping errors between MARCATS surfaces in Laruelle et al. (2014) and ours are particularly important for low area regions and reach 50% as maximums. Thus, we estimate that the model-data comparison using area-integrated $FCO_2$ is not adequate with such large remapping errors. We plan to delete this part of the model-data comparison and the 2$^{nd}$ panel (b) of the Figure 6 when revising the manuscript.

Page 10 line 14: what do you mean by 'top two regions'?

EBC and WBC are similar, they are the top two (the two largest) regions in absorbing anthropogenic carbon but their behavior differs in terms of total $FCO_2$. We will reformulate the sentence in the revised manuscript to clarify.

Page 10 line 25 and further: It is important to state that (to my knowledge) only LA14 accounts for the sea-ice cover in the global estimates - this is an important effect on the quantification.

We will emphasize that the Laruelle et al. (2014) is the first and only study to provide coastal observational-based $FCO_2$ estimates at global scale taking into account the effect of sea ice. Remarks will be added both to section 2.4 (Evaluation dataset) and to section 4.1.1 (Total fluxes).

Page 11 line 5-10: I agree that the exclusion of the proximal zone in the model assessment should have an impact on the sign of the flux under pre-industrial conditions. But what about the effect of the initialisation (the value of the sink is not reported for the coastal ocean in 1850)? Stated differently, is the global coastal ocean in equilibrium at the onset of the simulations? Regarding the proximal zone, bays, estuaries, deltas, lagoons are indeed sources of $CO_2$ (see Laruelle et al., 2013 for the latest synthesis), banks should be too (a reference would be useful), but I am not sure about what is meant by ' marine wetlands '. If this refers to marshes and mangroves, they are then believe to be sinks for atm $CO_2$ (see Cai, 2011, Regnier et al., 2013, Bauer et al., 2013). Thus, clarification is required here.

Certainly, the coastal ocean is not at equilibrium when the simulation is initialized in 1870. This is an important point addressed in our previous responses.

"Marine wetlands" will be replaced by "salt marshes and mangroves" in the revised manuscript to clarify the sentence and "banks" will be added to the revised manuscript. Those regions are included in the proximal zone that is generally known as a carbon source although some parts of it may be sinks.

Page 11, section 4.1.2 first paragraph: I believe that regional scale studies have attempted an estimation of the anthropogenic $CO_2$ uptake in EBUS. If true, they should be included in the discussion.

We have searched but fail into find regional-scale studies that provide estimates of anthropogenic $CO_2$ uptake in EBUS regions. Estimates of anthropogenic carbon content in the water column have been published for instance for the California current region by Feely

et al. (2008) but we do not found any regional estimates of anthropogenic $CO_2$ uptake from the atmosphere.

Page 11 line 21-22: This sentence has to be rephrased as it implies that one modeling approach performs better than another. Please tone down.

We do not understand Referee #1's remark to tone down the sentence on line 21-22. Our affirmation that Wanninkhof et al. (2013) exploit coarse-resolution model and data is valid. According to our model, the extrapolation technique used by Wanninkhof et al. (2013) overestimates the anthropogenic carbon uptake of the coastal ocean. We plan to leave the sentence as is.

Page 12, section 4.2: the first two paragraphs on total fluxes should be merged with section 4.1.1 – Regarding the Amazon, what is the potential impact of assuming that all the DOC (a large flux) is transformed into DIC in this region? More generally, do you assume that this instantaneous transformation has no impact on alkalinity?

We choose to design the section 4.2 to highlight contrasts between the coastal and the open ocean whereas 4.1.1 is specifically focused on coastal total FCO2. As the first two paragraphs of the section 4.2 deal with coastal vs. open ocean comparison, we suggest letting the paragraphs as is. To clarify the aim of section 4.1, we will rename it as "Comparison with previous coastal estimates".

Our assumption of the instantaneous remineralisation of all land-derived DOC into DIC impacts natural $FCO_2$ (and total $FCO_2$) but has no effect on simulated anthropogenic $FCO_2$. One of the potential of this assumption would be a total $CO_2$ sink reduction, shown for the Amazon plume for instance. In our model, river alkalinity input is equal to the initial riverine DIC input. But when riverine DOC is remineralized to DIC, that does not affect simulated ocean alkalinity. This point will be clarified in the model description.

Page 12, section 4.2: the latitudinal trends in anthropogenic $CO_2$ fluxes are also very similar for the coastal and open ocean (Figure 5). This aspect needs to be discussed.

The latitudinal distributions of anthropogenic $CO_2$ fluxes are indeed similar between the coastal and open ocean. In particular, we note that this similarity is prominent in the Southern Ocean: Antarctic shelves and adjacent open ocean waters are very much alike. Following the Laruelle et al. (2014) definition of the Antarctic Shelves, the bathymetry of this coastal region is deeper than the other MARCATS regions. Its mean bathymetry is around 500 m against 160 m for the global coastal ocean. This mitigates the contrast between coastal and open ocean processes in the Southern Ocean. This explanation will be added to the revised manuscript.

Page 13, lines 12-19: The computation of Revelle factor values is interesting, but it is important to stress that (to my knowledge), a higher value for the global coastal ocean compared to the global ocean remains highly speculative as this has not been demonstrated from observational data. Also, - and this is an important point – the sentence ' That finding is consistent with the lower simulated specific fluxes into the coastal ocean ' is not convincing. At the end, the Revelle factor should influence the total fluxes (and not its anthropogenic component) for which the area-based estimates indicate significantly larger negative sinks than in the global ocean (fig.5), i.e. the opposite of the anthropogenic component fluxes.

We insist that the Revelle factor does affect the anthropogenic $CO_2$ flux (e.g., Sabine, 2004, Nature). This fact can be demonstrated with simple equilibrium calculations. For example, using CO2SYS-Matlab, if we increase the $xCO_2$ from 280 to 400 ppm in equilibrium with two surface water masses at 2°C and 20°C each having a total alkalinity of 2300 ueq $kg^{-1}$, the corresponding increases in $DIC_{ant}$ are 55 and 73 umol $kg^{-1}$ (while the Revelle factor increases by 2.2 and 1.3, respectively). Clearly the Revelle factor influences anthropogenic carbon uptake.

In the criticized sentence, we will add "anthropogenic carbon" so that its revision will read as follows: "That finding is consistent with the lower simulated specific fluxes of anthropogenic carbon into the coastal ocean". The differences between coastal and global ocean highlighted by the referee in Figure 5 are mainly due to natural fluxes.

Sabine et al. (2004)
http://dx.doi.org/10.1126/science.1097403

Page 13 lines 23-24: The chemical factors are presented as independent of the physical factors controlling the air-sea $CO_2$ exchange. However, based on the model construct, I feel that the higher Revelle factor for the coastal ocean precisely results from the physics of the coastal zone, with a progressive accumulation of DIC due to weaker cross-shelf export than $CO_2$ air-sea exchange.

At the beginning of the historical simulation, the Revelle factor is already 16% larger on average, for the coastal ocean relative to the global ocean. And at the end of the simulation it remains about the same, 17% larger. We agree though that cross-shelf transport is inadequate to allow the coastal ocean to take up as much $CO_2$ per unit area as for the global ocean average.

Figure 8: I assume that fluxes refer to total anthropogenic fluxes, i.e. organic plus inorganic carbon – please clarify.

Yes, the flux in Fig. 8 refers to anthropogenic flux. And that corresponds only to an inorganic flux. The model does not account for the anthropogenic perturbation to the organic carbon pool (see general reply).

Spelling

Page 1, Line 10-11: rephrase – this sentence is odd

Initial sentence:
Yet only 4.5% of that (0.10 Pg C $yr^{-1}$) is absorbed by the global coastal ocean, i.e., less than its 7.5% proportion of the global ocean surface area.

New sentence:
Yet only 0.1 Pg C $yr^{-1}$ is absorbed by the global coastal ocean. That represents 4.5% of the anthropogenic carbon uptake of the global ocean, less than the 7.5% proportion of coastal-to-global ocean surface areas.

Page 2 line 4: remove 2nd 'relative'

Done.

Page 4 line 25: not sure that 'model' can be used as a verb

In the revised manuscript, we will replace "explicitly models" with "explicitly accounts for".

Page 10 line 19: remove 'that'

OK.

Page 14 line 14: remove 'of'

Line 13: We point out a missing "carbon" after the word "anthropogenic". It will be corrected in the revised manuscript.

"offshore transport of carbon" is replaced by "offshore carbon transport".

Page 14 line 24-25: parenthesis wrongly placed

Fixed.

- ### Referee #2 comments

1 Summary

Bourgeois et al. use an eddy-permitting global ocean biogeochemistry model to investigate the relative contribution of the ocean margins to the uptake of anthropogenic $CO_2$ from the atmosphere. They find that these regions take up a disproportionally low amount of anthropogenic $CO_2$, i.e., only 4.5% relative to this regions areal contribution of 7.5%. The authors suggest that it is the limited degree to which the anthropogenic $CO_2$ is transported and mixed offshore that leads to this low uptake. These results are in stark contrast to earlier studies based primarily on a few point observations that suggested a very high coastal ocean uptake of $CO_2$ from the atmosphere.

Thanks for this nice feedback.

2 Evaluation

Understanding and predicting the future evolution of the oceanic sink for atmospheric $CO_2$ is of paramount importance for determining how much $CO_2$ we can emit in the coming decades without exceeding any climate target. While our confidence in the net exchange fluxes of $CO_2$ over the open ocean has increased substantially in recent years thanks to better observations and novel methods to interpret these data, our ability to constrain the fluxes along the continental margins has not increased commensurably. The existing observational constraints are still relatively weak and associated with sizeable uncertainties. By far the most extensive and detailed assessment to date by Laruelle et al. suggested a relatively small global net uptake of atmospheric $CO_2$, but their data-based approach did not permit them to separate this net flux into its "natural" and "anthropogenic" components.

Thus the study by Bourgeois et al. is much welcomed as it brings a consistent global perspective to the problem. Their model-based approach is far from perfect, owing to many issues ranging from its still coarse resolution to the limited of consideration of several processes that are of relevance in coastal systems (e.g., limited consideration of benthic processes), but by using - for the first time - an eddy permitting 3D model, it is a big step up from previous model based approaches that used highly simplified models. I particularly like that the authors spend a considerable amount of effort in order to assess their model's performance against observations and that they run two parallel simulations in order to determine the anthropogenic $CO_2$ uptake explicitly. The paper is overall well written, adequately illustrated and referenced. The discussion is generally thorough and the conclusions supported by the provided evidence including an appropriate consideration of the caveats. All in all, this is a very good study that is well suited for publication in Biogeosciences. However, before giving the green light, I would like the authors to consider my concerns regarding important model shortcomings.

- Model drift: The model's drift is quite substantial, and I am not entirely convinced that the authors have fully considered the implications when discussing their results. I think the drift is large enough that it cannot be ignored.

Please see our general response.

- Reflective bottom boundary: The model's lower boundary is assumed to "reflect" any settling organic matter back into the water column in remineralized form. Thus, this model does not incorporate any delayed response of remineralization nor any other benthic remineralization process of relevance such as benthic denitrification etc. This is a rather important omission, as these processes influence coastal biogeochemistry in many shallow marginal seas. I thus recommend to address and discuss this issue in somewhat more detail and to give it better consideration.

We agree the referee on the need to give better consideration on that point for the model-data comparison.

In the revised manuscript, we will better describe the version of the model used in this study: in particular, we will indicate that part of the settling organic material is indeed reflected back in the water column in remineralized form, whereas part of it is buried to compensate for the riverine input.

In the new version of PISCES (PISCES-v2, Aumont et al. GMD, 2015), to be used in subsequent studies, water-sediment interactions are considered using the meta-model of Middelburg et al. (1996), which allows computation of sediment denitrification. It also explicitly represents conservation of calcite in the sediment as a function of the saturation levels of the overlying waters.

In the revised manuscript, we will also discuss some shortcomings of such representation on our results, referring for example to the work of Krumins et al. (2013) and Soetaert et al. (2000).

Krumins et al. (2013)
http://dx.doi.org/10.5194/bg-10-371-2013
Soetaert et al. (2000)
http://dx.doi.org/10.1016/S0012-8252(00)00004-0

- Coastal-open ocean exchange: This issue is actually addressed a bit more than the other two in the current version, but I still consider it worthwhile to assess the implications of this shortcoming in more detail.

See general reply (1) above.

3 Recommendation

I recommend acceptance of this manuscript with minor revisions.

4 Minor comments

Abstract, line 10: "absorbed by the coastal ocean": I suggest to define also in the abstract how the authors define the "coastal" ocean.

We agree on the need to explain our definition of the coastal ocean in the abstract. We will add "Here we define the coastal zone as the continental shelf area, excluding the proximal zone" in the abstract of the revised manuscript.

p1, line 15: "the ocean naturally mitigates..." I am not sure why the authors use the expression "naturally" here. I suggest deleting it.

We agree. "Naturally" will be deleted.

p5, line 5, equation 2: The authors still use a coefficient of 0.30, while there are numerous studies that have shown that this coefficient needs to be lowered in order to close the oceanic C14 budget. It's time to change, no?

Referee #2 is certainly correct. In future work, we will switch to using the revisited value of the coefficient (0.25) from Wanninkhof (2014, Limnol. Oceangr.), which is the new standard being adopted for the Ocean Model Intercomparison Project (OMIP). Here, we estimate that our 20% larger coefficient will lead to errors in anthropogenic carbon uptake of about 2% based on the study by Sarmiento et al (1992), who showed that a doubled coefficient increased anthropogenic carbon uptake by only 10%.

p5, lines 17 to 25: By neglecting the oceanic uptake of anthropogenic CO2 over the period 1750 until 1870, the authors underestimate the total anthropogenic uptake by about 10% or so. This should be taken into consideration more explicitly. I also suggest to avoid the use of the term "preindustrial" when referring to 1870.

In the revised manuscript, we will clearly define what we mean by preindustrial, prior to 1870, but that that operational definition does indeed neglect small changes between 1750 and 1870, resulting in about a 10% underestimate in our results.

p6, line 14: "as compared to the estimate of natural carbon outgassing of 0.45 Pg C yr-1". I find this comparison confusing, since the two processes are clearly distinct. The first number is the drift of the model, while the second one refers to the outgassing of river-derived carbon from the ocean. I do not think that these two numbers can be compared as done here.

Our integrated air-sea flux of -0.33 Pg C/yr is actually comparable to the 0.45 Pg C yr$^{-1}$ outgassing because at equilibrium the model should indeed have the same value as the latter. That is, its delivery of riverine carbon to the ocean minus sedimentary burial is the same number. We will clarify this point in the revised manuscript.

p8, line 14: "Regionally, [the] overall patterns in the air-sea $CO_2$ flux are similar between the ..." How was the large model drift considered when making this statement? This issue is mentioned further down (line 24), but not really elaborated.

We think that the general patterns in the $FCO_2$ would remain similar after a long spin up, except for the Southern Ocean where the drift is concentrated. Please see our general reply for more details.

p9, lines 16-17: "Correlation [..] only 0.5 for area-integrated fluxes". How important is the remapping error here? (see my comment on Table 2).

Remapping errors are particularly important for low area regions and reach 50% as maximums. Thus, following this comment and the next ones on Table 2 and Figure 6b, we agree Referee #2 that the model-data comparison using area-integrated $FCO_2$ is not adequate with such large remapping errors. We plan to delete this part and the 2$^{nd}$ panel (b) of the Figure 6 when revising the manuscript.

p11, lines 23-24 ".. carbon sink in the Amazon river plume [..] is not reproduced". I think that this also has something to do with the model's shortcoming with regard to the benthic-water column interactions.

Referee #2 offers a plausible explanation for this discrepancy. Another cause for discrepancy might be the instantaneous remineralisation of the entire land-derived DOC into DIC. We will mention these possibilities in the revised manuscript.

p14, lines 1-3: "... is transported offshore to the deeper open ocean". This deserves a little more discussion, particularly since a few lines below (e.g., lines 16-17) this issue is identified as "a critical question".

We agree. In the revised manuscript, there will be much new discussion on the subject offshore transport, as indicated in our general reply above.

Table 2: some of the areal discrepancies are huge. I thus think that one needs to be very careful when comparing areally integrated fluxes. In fact, this should not be done, in my opinion, if the areas differ by more than let's say 10%.

Good point. We agree. Please see our comment below regarding Fig. 6, for which we plan to delete the 2$^{nd}$ panel (b) when revising the manuscript.

Figure 2: reduction in uptake during the 1940s. Where is this coming from? To me, this looks like the model is overly sensitive to the rate of change in atm. $CO_2$, perhaps due to the drift.

Because the simulated ocean uptake of anthropogenic carbon is estimated from the difference of our 2 simulations, it bears only a very weak signature of climate (interannual, decadal) variability. The drop of uptake in the 1940s is hence clearly coming from the pause of atmospheric $CO_2$ growth during that period. We have checked this simulated drop by comparing it to all CMIP5 historical simulations, based on the recent paper from Bastos et al. (2016) in *Biogeosciences Discussions* in which the authors focus on the 1940s $CO_2$ plateau. The figure below indicates that our simulation is broadly consistent with the CMIP5 coupled simulations in terms of its response to the stall in atmospheric $CO_2$ during the 1940s (see also Figure 4 from Bastos et al., 2016). As indicated in Bastos et al. (2016), "The anomalies in ocean $CO_2$ uptake present multi-decadal variations which are consistent between the 16 models and are due to the ocean response to the atmospheric $CO_2$ forcing. In particular, during the plateau of the 1940s, most models estimate lower ocean uptake because of the slow-down of the anthropogenic perturbation".

[Figure]

*Figure 3: Ocean uptake of anthropogenic $CO_2$ simulated by the group of CMIP5 models and comparison with our model results (black line).*

Bastos, A., Ciais, P., Barichivitch, J., Bopp, L., Brovkin, V., Gasser, T., Peng, S., Pongratz, J., Viovy, N., and Trudinger, C.M. (2016). Re-evaluating the 1940s $CO_2$ plateau. Biogeosciences Discussions 1–35 (http://www.biogeosciences-discuss.net/bg-2016-171/)

Figure 6: See also my comment about Table 2. I am not sure whether it is really appropriate to compare the areally integrated fluxes when the areas are that different to begin with.

We agree. We will remove Figure 6b from the revised manuscript.

**Here are some that we propose to improve the manuscript, independent from the reviewers' comments**

- Update Figs. 3d and 5 of the submitted manuscript to remove previously unnoticed errors in zonal mean calculations.

- Update Fig. 2 of the submitted manuscript by replacing time series for global ocean to time series for open ocean (see the updated figure at the end)

- Add Jens Terhaar to the author list as he provided the residence-time calculation.

- Reduce file size and enhance of figure details for Figs. 3a, 3b, 3c, 3e, 3f, and 4 of the submitted manuscript.

- Add following link http://katirg.github.io/CO2airsea/ to permit access to a data visualization to that nicely illustrates results.

[revised manuscript text omitted]

---

## Author Response (AR1)

We thank Prof. Pierre Regnier (referee #1) and Prof. Nicolas Gruber (referee #2) for their comments and suggestions. We also thank Prof. Jack Middleburg for managing the peer-reviewing process of this manuscript. We first present our Author' Response with a point-by-point response to the reviews with relevant changes made in the revised manuscript, with referee comments in grey and author responses in black. Then we present the marked-up manuscript with additions and changes in **bold** and removals in **bold strikethrough**.

**Point-by-point response to the reviews with relevant changes made in the manuscript**

We first comment on 3 key topics raised by the referees, and then provide point-by-point responses for each referee with changes made in the manuscript. The 3 key topics are (1) the need for more information to support our proposed mechanism for lower coastal anthropogenic $CO_2$ uptake due to limitation by cross-shelf exchange, (2) the need to clarify our estimate of anthropogenic carbon fluxes and on the role of potential biological changes, and (3) the effect of model drift on our results.

**1) Cross-shelf exchange**

We concur with both referees that more information is needed to support our conclusion that it is inadequate cross-shelf exchange that reduces the coastal ocean's uptake of anthropogenic CO2 (per unit area) relative to that of the global ocean.

Following the suggestion by Referee #1, we plan to include a new figure (Figure 10) with time series of dissolved inorganic anthropogenic carbon ($DIC_{ant}$) storage and anthropogenic $CO_2$ uptake with cross-shelf export of $DIC_{ant}$. This new figure demonstrates clearly that simulated cross-shelf export of $DIC_{ant}$ is less than the simulated anthropogenic $CO_2$ uptake for the coastal ocean, implying an accumulation of $DIC_{ant}$ in the coastal waters column during the simulation.

As suggested by the Referee #2, we offer more detail about the simulated coastal-ocean ocean exchange. For that, we added the simulated water residence time for each MARCATS region to Tables 2 and 3 of the revised manuscript and we introduced a new figure (Figure 8). These new results reveal that simulated residence times for most coastal regions are of the order of a few months or less, except for Hudson Bay, the Baltic Sea and the Persian Gulf. The latter three regions are generally more confined and we would expect longer residence times, although our model simulations were never designed to simulate these regions accurately. Generally then, our simulated residence times are shorter than what has been published for similarly defined coastal regions (see Jickells et al. (1998), Men et al. (2015)). The absence of tides and the spatial resolution used in our study do not permit to reproduce the mesoscale variability of the coastal circulation (e.g. eddies and upwellings). Depending on coastal regions, these processes may increase/decrease residence time. The comparison with these published estimates is also difficult since we average residence time estimates on extended areas.

Page 9 Line 8: We added the section 2.6 in the Methods section to explain the residence time calculation:
"2.6 Residence time

To compute water residence time in each MARCATS region, we divided the volume of each region by the integrated outflow of water from 5-day mean current velocities at coastal boundaries."

Page 15 Line 19: We discussed simulated residence time as follows:
"The accumulation in the coastal ocean is effective over the entire period (1910-2012) as the uptake of anthropogenic carbon by the global coastal ocean is always inferior to its cross-shelf export (Fig. 10). To gain insight into this cross-shelf exchange, we computed the simulated mean water residence times for each MARCATS region (Fig. 8). Residence times for most coastal regions are of the order of a few months or less, except for Hudson Bay, the Baltic Sea and the Persian Gulf. The latter three regions are generally more confined and we expect longer residence times, although our model simulations were never designed to simulate these regions accurately. Generally, our simulated residence times are shorter than what has been published for similarly defined coastal regions although methods differ substantially (Jickells, 1998; Men and Liu, 2014; Delhez et al., 2004). Despite these short residence times, the cross-shelf export of anthropogenic carbon is unable to keep up with the increasing air-sea flux of anthropogenic carbon (Fig. 10). This may be explained by the open-ocean waters that are imported to the coastal ocean being already charged with anthropogenic carbon, thus limiting further uptake in the coastal zone."

[Figure]

**Figure 8. Global distribution of simulated residence time (month) for the global coastal ocean segmented following Laruelle et al. (2013).**

[Figure]

**Figure 10.** Simulated temporal evolution of (a) coastal-ocean inventory of anthropogenic carbon given in Pg C and (b) anthropogenic $CO_2$ ($C_{ant}$) uptake by the global coastal ocean and global cross-shelf export of anthropogenic carbon ($DIC_{ant}$) given in $Pg\,C\,yr^{-1}$.

**2) Our estimate of anthropogenic carbon fluxes and the role of potential biological changes**

Our method to compute anthropogenic air-sea $CO_2$ fluxes ($FCO_2$) is based on the difference between total $FCO_2$ from an historical simulation (with increasing atmospheric $CO_2$) and natural $FCO_2$ from a control simulation (i.e., with the same physical forcing but with fixed pre-industrial atmospheric $CO_2$). This method is commonly used by the modeling community to estimate anthropogenic carbon fluxes (e.g. http://ocmip5.ipsl.jussieu.fr/OCMIP/ or Bopp et al., 2015). By definition, our anthropogenic air-sea $CO_2$ fluxes only respond to increasing atmospheric $CO_2$. They do not include any effect from potential changes in ocean physics or biology, because those changes are identical in the historical and the control simulations. Hence, even if surface temperature or biological fluxes (NEP or NEC) change in response to the forcing, they do not impact anthropogenic carbon uptake per se.

That said, we agree with the reviewers that potential changes in the physics and biology as well as changes in riverine input or in the interactions with the sediment may be of primary importance, would modify the distribution of total carbon and alkalinity, and hence would also modify the potential of the coastal ocean to absorb anthropogenic carbon.

These points are discussed in more detail in the revised manuscript.

Page 2 Line 13: We added "Indeed, in addition to the effect of increasing atmospheric CO2, potential changes in coastal ocean physics (e. g., temperature) and biology (e.g., NEP) as well as changes in riverine input and interactions with the sediment may be of primary importance (Mackenzie et al., 2004; Hu and Cai, 2011). These changes would modify the distribution of carbon and alkalinity, and hence change the potential of the coastal ocean to absorb anthropogenic carbon."

To clarify our method at the end of the Introduction section:

Page 4 Line 4: We added "We focus solely on the geochemical effect of anthropogenic CO2 addition from the atmosphere to the ocean and neglect the role of varying river input and interactions with the sediment, as well as the feedback from a changing climate."

To clarifiy our method in the Methods section:

Page 6 Line 34: We added "Here, we use the conventional definition of anthropogenic carbon in the ocean used by previous global-ocean model studies (OCMIP, http://ocmip5.ipsl.jussieu.fr/OCMIP/ and e.g., Bopp et al. (2015)), namely that anthropogenic carbon comes only from the direct geochemical effect of increasing atmospheric CO2 and its subsequent invasion into the ocean. By definition, this anthropogenic FCO2 does not include any effect from potential changes in ocean physics or biology. In the model, there are no changes nor variability in riverine delivery of carbon and nutrients, and anthropogenic carbon is not buried in sediments."

Bopp et al. (2015)
http://dx.doi.org/10.1002/2015GL065073

**3) Model drift and potential implications.**

We concur with both reviewers that details about model drift are important. In the original manuscript we stated the following: "At lower resolution (ORCA2), after a spin up of 3000 years, there is 0.26 Pg C yr-1 greater globally integrated sea-to-air flux, relative to results after only a 50-year spin up. Nearly all of that enhanced sea-to-air CO2 flux due to the longer spin up comes from the Southern Ocean." Unfortunately, such spin-up length is currently out of reach regarding to computation costs of our ORCA05-PISCES configuration. We emphasize though that our anthropogenic $FCO_2$ estimates are expected to be influenced very little by model drift because of the way anthropogenic carbon is defined ($C_{ant} = C_{total} - C_{natural}$), i.e., drift affects both $C_{total}$ and $C_{natural}$ in the same way.

Page 7 Line 14: We added "Anthropogenic FCO2 estimates are expected to be influenced very little by model drift because of the way anthropogenic carbon is defined, i.e., drift affects both natural carbon and total carbon in the same way."

**POINT-BY-POINT REPLY**

- Referee #1 comments

The manuscript by Bourgeois et al. is the very first attempt to quantify the air-sea $CO_2$ flux for the global coastal ocean using a highly-resolved 3D model. The authors compare in a convincing way their model results with observational data and discuss in detail the obtained spatial variability in the air-water $CO_2$ exchange. The approach is well described and model results are solid; I am thus very supportive of this research. In addition, the authors have attempted a quantification of the anthropogenic perturbation on the air-sea $CO_2$ flux, with the key finding that the magnitude of the perturbation could be significantly smaller than previously taught. This is obviously an important result that further strengthens the value of this contribution. However, the latter aspect has several shortcomings that I believe need to be addressed fully (see in particular major comments 2) and 3) before publication.

Thanks for this positive general feedback.

Major comments

1) Uncertainties are only reported once for the anthropogenic $CO_2$ flux (0.1 +- 0.01). You need to explain how this uncertainty has been estimated. It is also much lower than the uncertainty on the total simulated flux (0.27 +- 0.07), which is quite surprising. More generally, uncertainties and their quantification method should be reported for all fluxes and consistently throughout the text.

The uncertainty on our estimate of the coastal-ocean uptake of anthropogenic carbon is defined as its interannual standard deviation over 1993-2012. This will be clarified to the revised manuscript; uncertainties are also given for all fluxes (see updated Tables 2 and 3 at the end of our response).

Since our uncertainty values are based on the interannual variability over 1993-2012, the uncertainty on the simulated total $FCO_2$ is much higher than that of the anthropogenic flux. That is, the strong variability of total $CO_2$ flux is due almost entirely to variability in its natural component so that the variability in their difference (anthropogenic flux) is small.

Another way to estimate uncertainties for such fluxes would be to use multiple models or sensitivity tests with one model where key parameters were varied. These exercises are currently out of reach and left for future work.

To clarify the definition used for uncertainties:

Page 8 Line 31: We formulate the sentence as is. "In addition, the model's uncertainty, computed as the interannual variability over 1993–2012, is compared to uncertainties in the observational estimates, computed as the standard deviation between flux parameterizations from Wanninkhof (1992) as modified by Takahashi et al. (2009), Ho et al. (2006) and Wanninkhof (1992)."

2) Section 4.1.2 on anthropogenic fluxes provides a suitable comparison with previous estimates. However, the last paragraph is misleading as one of the key reason why the size of the perturbation could be larger in Mackenzie and co-workers is the stimulation of the biological pump by enhanced land-derived nutrient inputs. These aspects should be included in the discussion, but also much earlier in the text (introduction and, eventually, title). That is, the authors should clearly state right from the start that they only consider atmospheric $CO_2$ as their sole anthropogenic driver. As a result, I believe that only the physical dissolution pump is impacted, i.e., the model should simulate constant net ecosystem productivity (NEP) and (I suspect) constant net ecosystem calcification (NEC) during the entire historical period. The values for NEP and NEC should be reported and discussed (a subject of intense debate within the coastal C community) as this could be (another) plausible reason for the discrepancy with earlier estimates. Finally, nothing is said about temperature effects on the uptake of $CO_2$. This aspect should also be included in the description/discussion.

We agree with the reviewer that the way we define the anthropogenic flux hampers any strict comparison with some previous estimates in which other terms are taken into account. This

will be clarified in the revised manuscript, and we refer the reviewer here only to our general reply above for a first response to this comment.

3) Section 4.2 provides an explanation for the smaller relative magnitude of the global coastal anthropogenic $CO_2$ uptake compared to the global ocean. As it is, Figs. 2, 5 and 8 do not satisfactorily substantiate the proposed mechanism. What are missing are plots of temporal evolution of (organic and inorganic) carbon accumulation (also % relative increase) and cross-shelf export for the entire simulation period. The Revelle factors should also be reported. Based on the proposed mechanism, I would suspect to see a progressive decrease of the ratio of anthropogenic carbon uptake of the coastal ocean to the global carbon uptake due to the accumulation of anthropogenic $CO_2$ in the coastal water column through time and this does not seem to be the case (Fig.2). I also would suspect to see a progressive increase in the Revelle factor (faster for the coastal ocean than open ocean) through time. In addition, it would be interesting to briefly discuss why the uptake fluxes per unit surface area for the shelf seem to be larger than for the open ocean under pre-industrial conditions (Fig. 5). Furthermore, the authors should report the calculated horizontal cross-shelf transport of water as this is a crucial number to sustain their conclusion (a first sensitivity analysis could have been useful in this context). Finally, I believe that comment 2 above (focus on the physical dissolution pump only) is also relevant in the context of section 4.2

We concur with the reviewer that more information is indeed needed to substantiate our proposed mechanism; we refer to the general reply (point 1, above) in which we indicate the additional figures and discussions that we introduced in the revised manuscript.

We have also calculated the evolution of the Revelle factor for the coastal and open ocean over the historical period. While the coastal Revelle factor is 16% more than in the open ocean in 1870, that ratio evolves to being only 17% more for the last simulated decades (around 2000). Thus different rates of change of Revelle factor between the coastal and open ocean cannot solely explain the amplified anthropogenic carbon accumulation in the water column of the coastal ocean.

Finally, concerning the horizontal cross-shelf transport of water, we do not fully understand the suggestion of the referee. We have chosen to compute water residence times for each MARCATS. We refer the reviewers to our general reply for this comment.

Other comments

Abstract and conclusion: the authors should also summarize the main results on the total fluxes as this is the first time that a model-data comparison is performed with a physically-resolved model at the global scale.

As proposed, the main results on the model-data comparison of total fluxes will be added to the abstract and conclusion as follows "Evaluation of the simulated air-sea fluxes of total CO2 for 45 coastal regions gave a correlation coefficient R of 0.8 when compared to observation-based estimates."

Abstract page 1, line 8: a high resolution is required not only to resolve the bathymetry, but also the complex coastal currents (which in my opinion are not all induced by the bathymetry)

"To begin to better resolve coastal bathymetry" is replaced by "to better resolve coastal bathymetry and complex coastal currents" in the revised manuscript.

Page 2 line 4: I suspect that the word export refers to 'export production'. I would clarify because in the context of this paper, it could also refer to 'cross-shelf export'

As the intended meaning was "cross-shelf export", "carbon export" is replaced by "primary productivity, export production and carbon burial" to the revised manuscript to clarify this sentence.

Page 2 line 5: the carbon export and burial fluxes are highly uncertain – see, e.g. Krumins et al. 2013 (BG) for a review. The same also holds for the productivity (even the sign of the NEP is uncertain – see Bauer et al., 2013. It is thus not correct to state that the air-sea $CO_2$ flux is the most uncertain of the C fluxes for the coastal ocean.

We presume that the referee refers to page 2, line 9 rather than line 5. We agree that the "less is known" formulation for coastal-ocean air-sea $CO_2$ exchange is not suitable here. In the revised manuscript, the last sentence of this paragraph is replaced by "All these estimates suffer from high uncertainties as do those for coastal-ocean air-sea $CO_2$ exchange (Laruelle et al., 2014), particularly its anthropogenic component."

Page 2 line 15-16: I agree about the $CO_2$ switch of the coastal ocean from source to sink, but do not agree fully with the proposed attribution. Mackenzie and co-workers highlight the change in NEP (from enhanced land nutrient inputs) as one of their key driver to explain the shift (see also Regnier et al., 2013 & Bauer et al., 2013 – for reviews). Please clarify (see also major comments).

Referee #1 does well to emphasize the need to introduce the proposed role of the changing NEP and riverine nutrient inputs in the $CO_2$ source-to-sink shift of the coastal ocean from Mackenzie et al. (2004). This proposed mechanism will be discussed in the revised manuscript as will be that from Bauer et al. (2013) to explain the $CO_2$ source-to-sink shift due only to the increased physical uptake of atmospheric $CO_2$ (with constant NEP).

To better introduce the next modifications:

Page 2 Line 25: We added "Therefore, estimates of anthropogenic carbon uptake by the global coastal ocean rely mainly on modelling, extrapolations from the open-ocean and/or closing- or/ balanced- budget approaches. An early modelling approach was proposed by Andersson and Mackenzie (2004) and Mackenzie et al. (2004)."

To take into account referee #1 comment:

Page Line 30: We added "They estimated that the preindustrial coastal ocean was a source of CO2 to the atmosphere and had recently or will switched to a CO2 sink. This source-to-sink switch is mainly caused by a shift in net ecosystem production (NEP) due to increased anthropogenic nutrient inputs (Andersson and Mackenzie, 2004; Mackenzie et al., 2004). Another proposed mechanism is simply linked to the anthropogenic increase in atmospheric CO2, considering constant NEP (Bauer et al., 2013)."

Page 3, line 20: I recommend making reference to the few published long time series of $CO_2$ observations (> 1 decade) for the coastal ocean. I agree nevertheless that these time series alone are sparse and short. Thus, an observation-based global extrapolation of the anthropogenic component is highly uncertain.

Two references dealing with long time series of $CO_2$ observations in the coastal ocean were added in the revised manuscript (Astor et al., 2013 with 1996-2008 $CO_2$ data at CARIACO station on Venezuelan coasts and Ishii et al., 2011 with the 1994-2008 $CO_2$ data along 137°E on Japanese coasts) to support the lack of data-based estimates. To our knowledge, these are the only available time series of $CO_2$ data of more than a decade.

Page 2, Line 20: We added "Estimating air-sea fluxes of anthropogenic CO2 in the coastal ocean would require multidecadal time-series of coastal CO2 observations in order to extract an anthropogenic signal from the strong coastal natural variability. Such time-series are still rare and probably not long enough. To our knowledge, the only available equivalent time-series are the Ishii et al. (2011) 1994-2008 time series along 137°E on Japanese coasts and the Astor et al. (2013) 1996-2008 time-series at the CARIACO station on Venezuelan coasts."

Astor et al. (2013)
http://dx.doi.org/10.1016/j.dsr2.2013.01.002
Ishii et al. (2011)
http://dx.doi.org/10.1029/2010JC006831

Page 4 line 24: calcite particles are included. This is not a satisfactory description. Please state clearly if your model accounts for calcification as this process has a potentially important impact on the air-sea $CO_2$ exchange (see also major comments).

We have chosen not to provide great detail on the model's calcification-related processes, which are described extensively elsewhere (Aumont and Bopp, 2006). The simulated NEC remains constant at a first order and does not respond to increasing anthropogenic $CO_2$. Moreover, calcification is identical in our historical and control simulations; hence, it does not affect the simulated anthropogenic carbon perturbation.

Page 4 line 29: do you mean atmospheric deposition?

Yes, "budgets" is replaced by "deposition" in the revised manuscript.

Page 6 line 4: Assuming that land derived DOC is entirely labile is a strong assumption. The flux (0.15 PgC yr-1 from the top of my head) is also significant. Thus, the extent to which your results depend on this assumption has to be discussed.

Referee #1 is absolutely right that the representation of riverine DOC input in the model is extremely simple. Dissolved organic matter is indeed assumed to remineralize instantaneously at the river mouths, thus contributing to the DIN, DIP, DIC pools. Yet complexifying the approach will not affect our estimate of anthropogenic carbon uptake since the land-derived carbon delivery and its lability remains constant throughout simulation. We agree with the referee on the need to clarify this point in the revised manuscript.

Page 6 lines 11-19: The implication of a model outside of ' equilibrium ' has to be addressed. For instance, when you refer to a global ocean anthropogenic uptake of 2.3 PgC/yr-1, this number is obtained with a natural flux of -0,33 PgC yr-1 for the natural flux. Correct?

This is correct. We refer the reviewers to the general reply for this comment.

Page 7, section 2.4 evaluation dataset: To leave no ambiguity, did you compare your model results with LA14 using the Wanninkhoff 1992 formulation or the updated formulation?

We used Laruelle et al. (2014) estimates computed using the formulation of Wanninkhof (1992) as modified by Takahashi et al. (2009) to compare with simulated $CO_2$ fluxes computed using the initial Wanninkhof et al. (1992) formulation. Although the gas exchange formulation is critical for air-sea flux estimates derived from observations, it has little impact on model-derived fluxes of anthropogenic carbon (Sarmiento et al., 1992).

Sarmiento et al. (1992)
http://dx.doi.org/10.1029/91JC02849

Page 9, lines 12-14: I would say 'weak carbon sources' and 'strong carbon sinks'

We made both of these changes in the revised manuscript.

Page 9 lines 15-25: The phrasing is misleading ('our model results tend to underestimate total carbon flux, with 76% of the simulated specific fluxes lower than the data-based estimates'), as the absolute fluxes are actually larger in the model (i.e. larger negative sinks). 'Likewise' is also not appropriate because the Arctic region is in fact the only latitudinal band where the model results predict a smaller sink than the observations. More broadly, I find that the results are quite comparable for the southern hemisphere and the low latitude regions, but that discrepancies are significantly larger in the Northern hemisphere with a stronger sink modeled for the 30-60◦ N and a weaker sink modeled for the > 60◦N latitudinal band (see also Fig 3 of LA14). Also, the fact that the areal-integrated fluxes show a weaker obs-model correlation than the fluxes per unit surface area requires discussion.

The sentence "our model results tend to underestimate total carbon flux, with 76% of the simulated specific fluxes lower than the data-based estimates" is replaced by "our model results tend to simulate larger sinks and weaker sources than observed (i.e. 76% of the specific simulated fluxes of total carbon have lower relative values than the data-based estimates)".

"Likewise" is replaced by "Otherwise".

Using the latitudinal distribution of natural fluxes, we can explain why carbon uptake is indeed larger in the coastal ocean. This is because upwelling zones (natural $CO_2$ sources) are extremely restricted (i.e. narrow continental shelf) contrary to the $CO_2$ sink regions with commonly large continental shelves.

Concerning the weak obs-model correlation for area-integrated fluxes, remapping errors between MARCATS surfaces in Laruelle et al. (2014) and ours are particularly important for low area regions and reach 50% as maximums. Thus, we estimate that the model-data comparison using area-integrated $FCO_2$ is not adequate with such large remapping errors. We

removed this part of the model-data comparison and the 2nd panel (b) of the Figure 6 in the revised manuscript.

Page 10 Line 18: We removed "but only 0.5 for area-integrated fluxes."

We replaced "top two regions" by "the most efficient regions in anthropogenic carbon uptake".

Page 10 line 25 and further: It is important to state that (to my knowledge) only LA14 accounts for the sea-ice cover in the global estimates - this is an important effect on the quantification.

We emphasized that the Laruelle et al. (2014) is the first and only study to provide coastal observational-based $FCO_2$ estimates at global scale taking into account the effect of sea ice. Remarks are added both to section 2.4 (Evaluation dataset) and to section 4.1.1 (Total fluxes) as follows.

Page 8 Line 24: we added "LA14 is the first and only study to provide coastal-ocean observation-based FCO2 estimates at global scale taking into"

Page 12 Line 2: We added "LA14 is the first observation-based study to take into account this sea-ice effect for coastal-ocean FCO2 estimates at global scale"

Page 11 line 5-10: I agree that the exclusion of the proximal zone in the model assessment should have an impact on the sign of the flux under pre-industrial conditions. But what about the effect of the initialisation (the value of the sink is not reported for the coastal ocean in 1850)? Stated differently, is the global coastal ocean in equilibrium at the onset of the simulations? Regarding the proximal zone, bays, estuaries, deltas, lagoons are indeed sources of $CO_2$ (see Laruelle et al., 2013 for the latest synthesis), banks should be too (a reference would be useful), but I am not sure about what is meant by ' marine wetlands '. If this refers to marshes and mangroves, they are then believe to be sinks for atm $CO_2$ (see Cai, 2011, Regnier et al., 2013, Bauer et al., 2013). Thus, clarification is required here.

Certainly, the coastal ocean is not at equilibrium when the simulation is initialized in 1870. This is an important point addressed in our previous responses.

"Marine wetlands" is replaced by "salt marshes and mangroves" in the revised manuscript to clarify the sentence and "banks" will be added to the revised manuscript. Those regions are included in the proximal zone that is generally known as a carbon source although some parts of it may be sinks.

Page 11, section 4.1.2 first paragraph: I believe that regional scale studies have attempted an estimation of the anthropogenic $CO_2$ uptake in EBUS. If true, they should be included in the discussion.

We have searched but fail into find regional-scale studies that provide estimates of anthropogenic $CO_2$ uptake in EBUS regions. Estimates of anthropogenic carbon content in the water column have been published for instance for the California current region by Feely

et al. (2008) but we do not found any regional estimates of anthropogenic $CO_2$ uptake from the atmosphere.

Page 11 line 21-22: This sentence has to be rephrased as it implies that one modeling approach performs better than another. Please tone down.

We do not understand Referee #1's remark to tone down the sentence on line 21-22. Our affirmation that Wanninkhof et al. (2013) exploit coarse-resolution model and data is valid. According to our model, the extrapolation technique used by Wanninkhof et al. (2013) overestimates the anthropogenic carbon uptake of the coastal ocean. We left the sentence as is.

Page 12, section 4.2: the first two paragraphs on total fluxes should be merged with section 4.1.1 – Regarding the Amazon, what is the potential impact of assuming that all the DOC (a large flux) is transformed into DIC in this region? More generally, do you assume that this instantaneous transformation has no impact on alkalinity?

We chose to design the section 4.2 to highlight contrasts between the coastal and the open ocean whereas 4.1.1 is specifically focused on coastal total FCO2. As the first two paragraphs of the section 4.2 deal with coastal vs. open ocean comparison, we suggest letting the paragraphs as is. To clarify the aim of section 4.1, we renamed it as "Comparison with previous coastal estimates".

Our assumption of the instantaneous remineralisation of all land-derived DOC into DIC impacts natural $FCO_2$ (and total $FCO_2$) but has no effect on simulated anthropogenic $FCO_2$. One of the potential of this assumption would be a total $CO_2$ sink reduction, shown for the Amazon plume for instance. In our model, river alkalinity input is equal to the initial riverine DIC input. But when riverine DOC is remineralized to DIC, that does not affect simulated ocean alkalinity. This point is clarified in the model description.

Page 12 Line 15: We added "The model representation of riverine DOC input and its instantaneous remineralization has potential implications for our estimates of total FCO2. In the Amazon plume for instance, we underestimate CO2 absorption because of this instantaneous addition of DIC without input of alkalinity. However this assumption has no direct implication on our anthropogenic FCO2 estimates."

Page 12, section 4.2: the latitudinal trends in anthropogenic $CO_2$ fluxes are also very similar for the coastal and open ocean (Figure 5). This aspect needs to be discussed.

The latitudinal distributions of anthropogenic $CO_2$ fluxes are indeed similar between the coastal and open ocean. In particular, we note that this similarity is prominent in the Southern Ocean: Antarctic shelves and adjacent open ocean waters are very much alike. Following the Laruelle et al. (2014) definition of the Antarctic Shelves, the bathymetry of this coastal region is deeper than the other MARCATS regions. Its mean bathymetry is around 500 m against 160 m for the global coastal ocean. This mitigates the contrast between coastal and open ocean processes in the Southern Ocean. This explanation is added to the revised manuscript.

Page 13 Line 33: We added "Yet the pattern for anthropogenic CO2 flux differs greatly from that of natural CO2, having its strongest uptake in the Southern Ocean in both the open and coastal oceans, i.e., where zonally averaged specific uptake reaches up to 1.5 molCm$^{-2}$ yr$^{-1}$.

The bathymetry of MARCATS regions around the Antarctic continent is much deeper than in the other coastal regions (500 m vs. 160 m for the global coastal ocean); this probably reduces the contrast between the coastal and open ocean in the Southern Ocean and explains the similarities of anthropogenic carbon uptake rates there."

Page 13, lines 12-19: The computation of Revelle factor values is interesting, but it is important to stress that (to my knowledge), a higher value for the global coastal ocean compared to the global ocean remains highly speculative as this has not been demonstrated from observational data. Also, - and this is an important point – the sentence ' That finding is consistent with the lower simulated specific fluxes into the coastal ocean ' is not convincing. At the end, the Revelle factor should influence the total fluxes (and not its anthropogenic component) for which the area-based estimates indicate significantly larger negative sinks than in the global ocean (fig.5), i.e. the opposite of the anthropogenic component fluxes.

We insist that the Revelle factor does affect the anthropogenic $CO_2$ flux (e.g., Sabine, 2004, Nature). This fact can be demonstrated with simple equilibrium calculations. For example, using CO2SYS-Matlab, if we increase the $xCO_2$ from 280 to 400 ppm in equilibrium with two surface water masses at 2°C and 20°C each having a total alkalinity of 2300 ueq kg$^{-1}$, the corresponding increases in $DIC_{ant}$ are 55 and 73 umol kg$^{-1}$ (while the Revelle factor increases by 2.2 and 1.3, respectively). Clearly the Revelle factor influences anthropogenic carbon uptake.

In the criticized sentence, we added "anthropogenic carbon" so that its revision is as follows: "That finding is consistent with the lower simulated specific fluxes of anthropogenic carbon into the coastal ocean". The differences between coastal and global ocean highlighted by the referee in Figure 5 are mainly due to natural fluxes.

Sabine et al. (2004)
http://dx.doi.org/10.1126/science.1097403

Page 13 lines 23-24: The chemical factors are presented as independent of the physical factors controlling the air-sea $CO_2$ exchange. However, based on the model construct, I feel that the higher Revelle factor for the coastal ocean precisely results from the physics of the coastal zone, with a progressive accumulation of DIC due to weaker cross-shelf export than $CO_2$ air-sea exchange.

At the beginning of the historical simulation, the Revelle factor is already 16% larger on average, for the coastal ocean relative to the global ocean. And at the end of the simulation it remains about the same, 17% larger. We agree though that cross-shelf transport is inadequate to allow the coastal ocean to take up as much $CO_2$ per unit area as for the global ocean average.

Figure 8: I assume that fluxes refer to total anthropogenic fluxes, i.e. organic plus inorganic carbon – please clarify.

Yes, the flux in Fig. 8 refers to anthropogenic flux. And that corresponds only to an inorganic flux. The model does not account for the anthropogenic perturbation to the organic carbon pool (see general reply).

Spelling

Page 1, Line 10-11: rephrase – this sentence is odd

Initial sentence:
Yet only 4.5% of that (0.10 Pg C yr$^{-1}$) is absorbed by the global coastal ocean, i.e., less than its 7.5% proportion of the global ocean surface area.

New sentence:
Yet only 0.1 Pg C yr$^{-1}$ of that is absorbed by the global coastal ocean. That represents 4.5% of the anthropogenic carbon uptake of the global ocean, less than the 7.5% proportion of coastal-to-global ocean surface areas.

Page 2 line 4: remove 2nd 'relative'

Done.

Page 4 line 25: not sure that 'model' can be used as a verb

In the revised manuscript, we replaced "explicitly models" with "explicitly accounts for".

Page 10 line 19: remove 'that'

OK.

Page 14 line 14: remove 'of'

Line 13: We point out a missing "carbon" after the word "anthropogenic". It will be corrected in the revised manuscript.

"offshore transport of carbon" is replaced by "offshore carbon transport".

Page 14 line 24-25: parenthesis wrongly placed

Fixed.

- Referee #2 comments

1 Summary

Bourgeois et al. use an eddy-permitting global ocean biogeochemistry model to investigate the relative contribution of the ocean margins to the uptake of anthropogenic $CO_2$ from the atmosphere. They find that these regions take up a disproportionally low amount of anthropogenic $CO_2$, i.e., only 4.5% relative to this regions areal contribution of 7.5%. The authors suggest that it is the limited degree to which the anthropogenic $CO_2$ is transported and mixed offshore that leads to this low uptake. These results are in stark contrast to earlier studies based primarily on a few point observations that suggested a very high coastal ocean uptake of $CO_2$ from the atmosphere.

Thanks for this nice feedback.

2 Evaluation

Understanding and predicting the future evolution of the oceanic sink for atmospheric $CO_2$ is of paramount importance for determining how much $CO_2$ we can emit in the coming decades without exceeding any climate target. While our confidence in the net exchange fluxes of $CO_2$ over the open ocean has increased substantially in recent years thanks to better observations and novel methods to interpret these data, our ability to constrain the fluxes along the continental margins has not increased commensurably. The existing observational constraints are still relatively weak and associated with sizeable uncertainties. By far the most extensive and detailed assessment to date by Laruelle et al. suggested a relatively small global net uptake of atmospheric $CO_2$, but their data-based approach did not permit them to separate this net flux into its "natural" and "anthropogenic" components.

Thus the study by Bourgeois et al. is much welcomed as it brings a consistent global perspective to the problem. Their model-based approach is far from perfect, owing to many issues ranging from its still coarse resolution to the limited of consideration of several processes that are of relevance in coastal systems (e.g., limited consideration of benthic processes), but by using - for the first time - an eddy permitting 3D model, it is a big step up from previous model based approaches that used highly simplified models. I particularly like that the authors spend a considerable amount of effort in order to assess their model's performance against observations and that they run two parallel simulations in order to determine the anthropogenic $CO_2$ uptake explicitly. The paper is overall well written, adequately illustrated and referenced. The discussion is generally thorough and the conclusions supported by the provided evidence including an appropriate consideration of the caveats. All in all, this is a very good study that is well suited for publication in Biogeosciences. However, before giving the green light, I would like the authors to consider my concerns regarding important model shortcomings.

- Model drift: The model's drift is quite substantial, and I am not entirely convinced that the authors have fully considered the implications when discussing their results. I think the drift is large enough that it cannot be ignored.

Please see our general response.

- Reflective bottom boundary: The model's lower boundary is assumed to "reflect" any settling organic matter back into the water column in remineralized form. Thus, this model does not incorporate any delayed response of remineralization nor any other benthic remineralization process of relevance such as benthic denitrification etc. This is a rather important omission, as these processes influence coastal biogeochemistry in many shallow marginal seas. I thus recommend to address and discuss this issue in somewhat more detail and to give it better consideration.

We agree the referee on the need to give better consideration on that point for the model-data comparison.

In the revised manuscript, we better described the version of the model used in this study: in particular, we indicated that part of the settling organic material is indeed reflected back in the water column in remineralized form, whereas part of it is buried to compensate for the riverine input.

Page 5 Line 15: We added "Those burial rates are hence dependent on the local sinking fluxes, but are set to balance inputs from rivers and atmospheric deposition at the global scale."

In the new version of PISCES (PISCES-v2, Aumont et al. GMD, 2015), to be used in subsequent studies, water-sediment interactions are considered using the meta-model of Middelburg et al. (1996), which allows computation of sediment denitrification. It also explicitly represents conservation of calcite in the sediment as a function of the saturation levels of the overlying waters.

In the revised manuscript, we discussed some shortcomings of such representation on our results, referring for example to the work of Krumins et al. (2013) and Soetaert et al. (2000).

Page 12 Line 19: We added "Furthermore, our simplified representation of sedimentary processes affects simulated total CO2 fluxes (Krumins et al., 2013; Soetaert et al., 2000). First, the model lacks an explicit representation of sedimentary processes. Thus it cannot reproduce the temporal dynamics of interactions between sediments and the overlying water column, e.g., resulting in potential delays between sediment burial and remineralization. Second, our model neglects any alkalinity source from sediment anaerobic degradation, such as denitrification and sulfate reduction of deposited organic matter. Even if not well constrained (Chen, 2002; Thomas et al., 2009; Hu and Cai, 2011; Krumins et al., 2013), this source of alkalinity could partially balance the total CO2 uptake of the coastal ocean. However, the simplified representation of these sediment processes has no direct effect on our anthropogenic FCO2 estimates."

Krumins et al. (2013)
http://dx.doi.org/10.5194/bg-10-371-2013
Soetaert et al. (2000)
http://dx.doi.org/10.1016/S0012-8252(00)00004-0

- Coastal-open ocean exchange: This issue is actually addressed a bit more than the other two in the current version, but I still consider it worthwhile to assess the implications of this shortcoming in more detail.

See general reply (1) above.

3 Recommendation

I recommend acceptance of this manuscript with minor revisions.

4 Minor comments

Abstract, line 10: "absorbed by the coastal ocean": I suggest to define also in the abstract how the authors define the "coastal" ocean.

We agree on the need to explain our definition of the coastal ocean in the abstract. We added "Here we define the coastal zone as the continental shelf area, excluding the proximal zone." in the abstract of the revised manuscript.

p1, line 15: "the ocean naturally mitigates..." I am not sure why the authors use the expression "naturally" here. I suggest deleting it.

We agree. "Naturally" is deleted.

p5, line 5, equation 2: The authors still use a coefficient of 0.30, while there are numerous studies that have shown that this coefficient needs to be lowered in order to close the oceanic C14 budget. It's time to change, no?

Referee #2 is certainly correct. In future work, we will switch to using the revisited value of the coefficient (0.25) from Wanninkhof (2014, Limnol. Oceangr.), which is the new standard being adopted for the Ocean Model Intercomparison Project (OMIP). Here, we estimate that our 20% larger coefficient will lead to errors in anthropogenic carbon uptake of about 2% based on the study by Sarmiento et al (1992), who showed that a doubled coefficient increased anthropogenic carbon uptake by only 10%.

p5, lines 17 to 25: By neglecting the oceanic uptake of anthropogenic CO2 over the period 1750 until 1870, the authors underestimate the total anthropogenic uptake by about 10% or so. This should be taken into consideration more explicitly. I also suggest to avoid the use of the term "preindustrial" when referring to 1870.

In the revised manuscript, we clearly defined what we mean by preindustrial, prior to 1870, but that that operational definition does indeed neglect small changes between 1750 and 1870.

Page 6 Line 12: We added "The preindustrial reference year is defined as 1870, thus neglecting changes in anthropogenic carbon storage in the ocean from 1750 to 1870."

p6, line 14: "as compared to the estimate of natural carbon outgassing of 0.45 Pg C yr-1". I find this comparison confusing, since the two processes are clearly distinct. The first number is the drift of the model, while the second one refers to the outgassing of river-derived carbon from the ocean. I do not think that these two numbers can be compared as done here.

Our integrated air-sea flux of -0.33 Pg C/yr is actually comparable to the 0.45 Pg C yr$^{-1}$ outgassing because at equilibrium the model should indeed have the same value as the latter. That is, its delivery of riverine carbon to the ocean minus sedimentary burial is the same number.

p8, line 14: "Regionally, [the] overall patterns in the air-sea $CO_2$ flux are similar between the ..." How was the large model drift considered when making this statement? This issue is mentioned further down (line 24), but not really elaborated.

We think that the general patterns in the $FCO_2$ would remain similar after a long spin up, except for the Southern Ocean where the drift is concentrated. Please see our general reply for more details.

p9, lines 16-17: "Correlation [..] only 0.5 for area-integrated fluxes". How important is the remapping error here? (see my comment on Table 2).

Remapping errors are particularly important for low area regions and reach 50% as maximums. Thus, following this comment and the next ones on Table 2 and Figure 6b, we agree Referee #2 that the model-data comparison using area-integrated $FCO_2$ is not adequate with such large remapping errors. We removed this part and the $2^{nd}$ panel (b) of the Figure 6 in the revised manuscript.

p11, lines 23-24 ".. carbon sink in the Amazon river plume [..] is not reproduced". I think that this also has something to do with the model's shortcoming with regard to the benthic-water column interactions.

Referee #2 offers a plausible explanation for this discrepancy. Another cause for discrepancy might be the instantaneous remineralisation of the entire land-derived DOC into DIC. We mentioned these possibilities in the revised manuscript.

Page 14 Line 15: We added "This discrepancy may be due to the modelled instantaneous remineralisation of land-derived DOC or to shortcomings in the model representation of sedimentary processes."

p14, lines 1-3: "... is transported offshore to the deeper open ocean". This deserves a little more discussion, particularly since a few lines below (e.g., lines 16-17) this issue is identified as "a critical question".

We agree. In the revised manuscript, there is much discussion on this subject, as indicated in our general reply above.

Table 2: some of the areal discrepancies are huge. I thus think that one needs to be very careful when comparing areally integrated fluxes. In fact, this should not be done, in my opinion, if the areas differ by more than let's say 10%.

Good point. We agree. Please see our comment below regarding Fig. 6, for which we removed the $2^{nd}$ panel (b) in the revised manuscript.

Figure 2: reduction in uptake during the 1940s. Where is this coming from? To me, this looks like the model is overly sensitive to the rate of change in atm. $CO_2$, perhaps due to the drift.

Because the simulated ocean uptake of anthropogenic carbon is estimated from the difference of our 2 simulations, it bears only a very weak signature of climate (interannual, decadal) variability. The drop of uptake in the 1940s is hence clearly coming from the pause of atmospheric $CO_2$ growth during that period. We have checked this simulated drop by comparing it to all CMIP5 historical simulations, based on the recent paper from Bastos et al. (2016) in *Biogeosciences Discussions* in which the authors focus on the 1940s $CO_2$ plateau. The figure below indicates that our simulation is broadly consistent with the CMIP5 coupled simulations in terms of its response to the stall in atmospheric $CO_2$ during the 1940s (see also Figure 4 from Bastos et al., 2016). As indicated in Bastos et al. (2016), "The anomalies in ocean $CO_2$ uptake present multi-decadal variations which are consistent between the 16 models and are due to the ocean response to the atmospheric $CO_2$ forcing. In particular, during the plateau of the 1940s, most models estimate lower ocean uptake because of the slow-down of the anthropogenic perturbation".

[Figure]

*Figure 1: Ocean uptake of anthropogenic $CO_2$ simulated by the group of CMIP5 models and comparison with our model results (black line).*

Bastos, A., Ciais, P., Barichivitch, J., Bopp, L., Brovkin, V., Gasser, T., Peng, S., Pongratz, J., Viovy, N., and Trudinger, C.M. (2016). Re-evaluating the 1940s $CO_2$ plateau. Biogeosciences Discussions 1−35 (http://www.biogeosciences-discuss.net/bg-2016-171/)

Figure 6: See also my comment about Table 2. I am not sure whether it is really appropriate to compare the areally integrated fluxes when the areas are that different to begin with.

We agree. We removed Figure 6b from the revised manuscript.

**Here are some improvements done in the revised manuscript, independent from the reviewers' comments**

- Add Jens Terhaar to the author list as he provided the residence-time calculation.

- Update Figs. 3d and 5 of the submitted manuscript to remove previously unnoticed errors in zonal mean calculations.

- Update Fig. 2 of the submitted manuscript by replacing time series for global ocean to time series for open ocean (see the updated figure at the end)

- Reduce file size and enhance of figure details for Figs. 3a, 3b, 3c, 3e, 3f, and 4 of the submitted manuscript.

- Figure 4 has been enriched with 2 additional panels. Panel b) is the same as Panel a) but for anthropogenic $CO_2$ fluxes. Panel c) is a bar chart illustrating anthropogenic carbon uptake from area-integrated fluxes for each MARCATS according to MARCATS class. Figure 4's label is completed with a link to a data visualization application (http://lsce-datavisgroup.github.io/CoastalCO2Flux/)

- We provided additional details at the end of Figure 6's label: "All MARCATS regions have been used except the Black Sea, the Persian Gulf (no data estimate), and the Sea of Okhotsk (see text)". The sentence "Corrections were also applied for the Florida-Labrador delimitation" is added for information in the marked-up manuscript but is removed in the revised manuscript. See next point for details about the Florida-Labrador delimitation.

- We noticed a mistake in the shapefile delimitation for the MARCATS Florida Upwelling and Labrador Sea. The Newfoundland was linked to the Florida Upwelling whereas Newfoundland is associated to the Labrador Sea in LA13. Thus, we updated flux estimates as well as Table 2, Table 3 and Figure 1, 4, 6, and 8.

- We forgot to remove the Sea of Okhotsk estimates for the computation of R correlation coefficient. As initially stated, this observation-based estimate "is not taken into account due to the extremely poor data coverage of this region and its strong divergence with the local literature (LA14)." We updated the R correlation coefficient to 0.8, instead of 0.7 initially. This improvement is also due to the Florida-Labrador correction stated earlier.

- Replace initial wrong reference for Revelle factor in the section 2.5 by Sundquist et al. (1979)

- To enhance sentence formulation, Page 12 Line 5 is reformulated as follows: "however, they also estimate that industrialisation has recently led to a reversal in the sign of this flux (the global coastal ocean became a carbon sink) mainly due to the enhancement of NEP from increased riverine inputs."

Please find below the marked-up manuscript with additions and changes in **bold** as well as removals in **bold strikethrough**.

[revised manuscript text omitted]
 | 397 | 350 | $-2.29 \pm 0.17$ | $-1.61$ | $-10.935 \pm 0.823$ | $-6.775$ | $-0.45 \pm 0.05$ | $-2.16 \pm 0.23$ | $0.83 \pm 0.23$ |
| 2 | Californian Current | EBC | 118 | 208 | $-0.34 \pm 0.10$ | $-0.05$ | $-0.477 \pm 0.148$ | $-0.135$ | $-0.35 \pm 0.09$ | $-0.50 \pm 0.13$ | $1.00 \pm 0.23$ |
| 3 | Tropical E Pacific | Tropical | 152 | 183 | $-0.12 \pm 0.05$ | $0.09$ | $-0.222 \pm 0.095$ | $0.192$ | $-0.36 \pm 0.05$ | $-0.65 \pm 0.10$ | $0.51 \pm 0.09$ |
| 4 | Peruvian Upwelling Current | EBC | 138 | 143 | $1.44 \pm 0.80$ | $0.65$ | $2.386 \pm 1.325$ | $1.073$ | $-0.39 \pm 0.09$ | $-0.64 \pm 0.15$ | $0.72 \pm 0.15$ |
| 5 | Southern America | Subpolar | 1126 | 1190 | $-1.51 \pm 0.13$ | $-1.31$ | $-20.460 \pm 1.705$ | $-18.715$ | $-0.46 \pm 0.05$ | $-6.28 \pm 0.74$ | $0.65 \pm 0.05$ |
| 6 | Brazilian Current | WBC | 475 | 484 | $-0.33 \pm 0.08$ | $0.10$ | $-1.872 \pm 0.479$ | $0.567$ | $-0.34 \pm 0.05$ | $-1.95 \pm 0.29$ | $0.26 \pm 0.06$ |
| 7 | Tropical W Atlantic | Tropical | 479 | 488 | $0.86 \pm 0.10$ | $0.07$ | $4.934 \pm 0.551$ | $0.394$ | $-0.26 \pm 0.05$ | $-1.50 \pm 0.31$ | $0.20 \pm 0.02$ |
| 8 | Caribbean Sea | Tropical | 303 | 358 | $0.10 \pm 0.10$ | $0.81$ | $0.366 \pm 0.348$ | $3.460$ | $-0.31 \pm 0.04$ | $-1.12 \pm 0.14$ | $0.32 \pm 0.03$ |
| 9 | Gulf of Mexico | Marginal Sea | 469 | 532 | $-0.79 \pm 0.11$ | $-0.33$ | $-4.478 \pm 0.633$ | $-2.100$ | $-0.32 \pm 0.03$ | $-1.81 \pm 0.16$ | $1.01 \pm 0.15$ |
| 10 | Florida Upwelling | WBC | 545 | 591 | $-2.25 \pm 0.21$ | $-0.38$ | $-14.692 \pm 1.351$ | $-2.723$ | $-0.66 \pm 0.05$ | $-4.29 \pm 0.36$ | $0.39 \pm 0.02$ |
| 11 | Sea of Labrador | Subpolar | 576 | 638 | $-1.27 \pm 0.18$ | $-1.72$ | $-8.808 \pm 1.244$ | $-13.172$ | $-0.32 \pm 0.03$ | $-2.19 \pm 0.21$ | $1.20 \pm 0.35$ |
| 12 | Hudson Bay | Marginal Sea | 998 | 1064 | $0.31 \pm 0.29$ | n.d | $3.757 \pm 3.423$ | n.d. | $-0.08 \pm 0.04$ | $-0.99 \pm 0.46$ | $51.22 \pm 22.75$ |
| 13 | Canadian Archipelago | Polar | 1001 | 1145 | $-0.52 \pm 0.06$ | $-1.02$ | $-6.234 \pm 0.748$ | $-13.986$ | $-0.09 \pm 0.02$ | $-1.03 \pm 0.21$ | $2.82 \pm 0.46$ |
| 14 | N Greenland | Polar | 544 | 602 | $-0.97 \pm 0.15$ | $-0.61$ | $-6.333 \pm 1.000$ | $-4.400$ | $-0.26 \pm 0.05$ | $-1.67 \pm 0.33$ | $2.38 \pm 0.44$ |
| 15 | S Greenland | Polar | 238 | 262 | $-3.35 \pm 0.44$ | $-3.81$ | $-9.564 \pm 1.259$ | $-11.972$ | $-0.86 \pm 0.19$ | $-2.45 \pm 0.53$ | $0.48 \pm 0.09$ |
| 16 | Norwegian Basin | Polar | 141 | 162 | $-2.87 \pm 0.23$ | $-1.72$ | $-4.855 \pm 0.396$ | $-3.342$ | $-0.60 \pm 0.09$ | $-1.02 \pm 0.15$ | $0.31 \pm 0.10$ |
| 17 | N-E Atlantic | Subpolar | 1020 | 1073 | $-2.16 \pm 0.12$ | $-1.33$ | $-26.501 \pm 1.419$ | $-17.165$ | $-0.53 \pm 0.05$ | $-6.52 \pm 0.59$ | $0.93 \pm 0.11$ |
| 18 | Baltic Sea | Marginal Sea | 324 | 364 | $0.30 \pm 0.07$ | $0.51$ | $1.184 \pm 0.288$ | $2.245$ | $-0.01 \pm 0.01$ | $-0.05 \pm 0.03$ | $17.37 \pm 9.52$ |
| 19 | Iberian Upwelling | EBC | 251 | 267 | $-1.13 \pm 0.12$ | $0.04$ | $-3.393 \pm 0.352$ | $0.122$ | $-0.27 \pm 0.03$ | $-0.82 \pm 0.10$ | $2.31 \pm 0.54$ |
| 20 | Mediterranean Sea | Marginal Sea | 423 | 529 | $-0.24 \pm 0.06$ | $0.62$ | $-1.196 \pm 0.327$ | $3.925$ | $-0.30 \pm 0.02$ | $-1.52 \pm 0.12$ | $0.72 \pm 0.09$ |
| 21 | Black Sea | Marginal Sea | 131 | 172 | $-0.24 \pm 0.11$ | n.d. | $-0.375 \pm 0.174$ | n.d. | $-0.18 \pm 0.02$ | $-0.28 \pm 0.03$ | $1.60 \pm 0.48$ |
| 22 | Moroccan Upwelling | EBC | 177 | 206 | $0.18 \pm 0.12$ | $2.92$ | $0.385 \pm 0.263$ | $7.220$ | $-0.33 \pm 0.03$ | $-0.71 \pm 0.07$ | $0.67 \pm 0.14$ |
| 23 | Tropical E Atlantic | Tropical | 225 | 259 | $0.09 \pm 0.08$ | $-0.06$ | $0.239 \pm 0.208$ | $-0.174$ | $-0.19 \pm 0.02$ | $-0.52 \pm 0.06$ | $0.59 \pm 0.09$ |
| 24 | S W Africa | EBC | 300 | 298 | $0.43 \pm 0.40$ | $-1.43$ | $1.544 \pm 1.448$ | $-5.103$ | $-0.59 \pm 0.08$ | $-2.14 \pm 0.28$ | $2.17 \pm 0.55$ |
| 25 | Agulhas Current | WBC | 189 | 239 | $-1.20 \pm 0.09$ | $-0.58$ | $-2.730 \pm 0.206$ | $-1.664$ | $-0.53 \pm 0.05$ | $-1.21 \pm 0.12$ | $0.13 \pm 0.01$ |
| 26 | Tropical W Indian | Tropical | 46 | 68 | $-0.06 \pm 0.08$ | $1.00$ | $-0.031 \pm 0.044$ | $0.815$ | $-0.16 \pm 0.04$ | $-0.09 \pm 0.03$ | $0.20 \pm 0.04$ |
| 27 | W Arabian Sea | Indian Margins | 82 | 92 | $0.35 \pm 0.04$ | $1.14$ | $0.342 \pm 0.043$ | $1.257$ | $-0.31 \pm 0.04$ | $-0.31 \pm 0.04$ | $0.12 \pm 0.04$ |
| 28 | Red Sea | Marginal Sea | 158 | 174 | $0.24 \pm 0.03$ | $0.16$ | $0.460 \pm 0.065$ | $0.330$ | $-0.15 \pm 0.01$ | $-0.28 \pm 0.02$ | $0.57 \pm 0.15$ |
| 29 | Persian Gulf | Marginal Sea | 208 | 233 | $0.04 \pm 0.08$ | n.d. | $0.092 \pm 0.203$ | n.d. | $-0.12 \pm 0.02$ | $-0.31 \pm 0.04$ | $24.67 \pm 12.09$ |
| 30 | E Arabian Sea | Indian Margins | 298 | 317 | $0.21 \pm 0.12$ | $0.67$ | $0.749 \pm 0.427$ | $2.555$ | $-0.30 \pm 0.04$ | $-1.07 \pm 0.19$ | $0.67 \pm 0.15$ |
| 31 | Bay of Bengal | Indian Margins | 197 | 203 | $-0.69 \pm 0.12$ | $-0.22$ | $-1.641 \pm 0.276$ | $-0.530$ | $-0.31 \pm 0.04$ | $-0.74 \pm 0.09$ | $0.43 \pm 0.11$ |
| 32 | Tropical E Indian | Indian Margins | 727 | 763 | $-0.06 \pm 0.07$ | $-0.02$ | $-0.482 \pm 0.569$ | $-0.170$ | $-0.20 \pm 0.02$ | $-1.78 \pm 0.17$ | $0.50 \pm 0.04$ |
| 33 | Leeuwin Current | EBC | 81 | 117 | $-2.05 \pm 0.15$ | $-0.98$ | $-2.010 \pm 0.148$ | $-1.379$ | $-0.60 \pm 0.07$ | $-0.58 \pm 0.07$ | $0.56 \pm 0.16$ |
| 34 | S Australia | Subpolar | 392 | 436 | $-1.37 \pm 0.18$ | $-1.14$ | $-6.438 \pm 0.859$ | $-5.983$ | $-0.27 \pm 0.03$ | $-1.29 \pm 0.14$ | $0.74 \pm 0.25$ |
| 35 | E Australian Current | WBC | 98 | 130 | $-1.74 \pm 0.18$ | $-1.09$ | $-2.036 \pm 0.205$ | $-1.695$ | $-0.50 \pm 0.07$ | $-0.58 \pm 0.08$ | $0.37 \pm 0.04$ |
| 36 | New Zealand | Subpolar | 263 | 286 | $-1.23 \pm 0.16$ | $-1.25$ | $-3.882 \pm 0.498$ | $-4.274$ | $-0.52 \pm 0.07$ | $-1.64 \pm 0.23$ | $0.49 \pm 0.04$ |
| 37 | N Australia | Tropical | 2278 | 2292 | $-0.29 \pm 0.11$ | $0.44$ | $-7.872 \pm 3.114$ | $12.120$ | $-0.23 \pm 0.04$ | $-6.19 \pm 1.00$ | $0.38 \pm 0.03$ |
| 38 | S-E Asia | Tropical | 2130 | 2160 | $-0.29 \pm 0.07$ | $-0.91$ | $-7.344 \pm 1.908$ | $-23.609$ | $-0.20 \pm 0.03$ | $-5.01 \pm 0.72$ | $0.49 \pm 0.05$ |
| 39 | China Sea and Kuroshio | WBC | 1132 | 1129 | $-1.99 \pm 0.18$ | $-1.41$ | $-27.046 \pm 1.991$ | $-19.100$ | $-0.45 \pm 0.05$ | $-6.13 \pm 0.72$ | $0.32 \pm 0.01$ |
| 40 | Sea of Japan | Marginal Sea | 233 | 147 | $-3.07 \pm 0.17$ | $-3.47$ | $-8.613 \pm 0.475$ | $-6.113$ | $-0.51 \pm 0.06$ | $-1.44 \pm 0.18$ | $1.64 \pm 0.24$ |
| 41 | Sea of Okhotsk | Marginal Sea | 933 | 952 | $-1.66 \pm 0.07$ | $1.31$ | $-18.623 \pm 0.761$ | $14.955$ | $-0.36 \pm 0.03$ | $-4.00 \pm 0.34$ | $3.52 \pm 1.38$ |
| 42 | N-W Pacific | Subpolar | 1025 | 1000 | $-1.85 \pm 0.14$ | $-0.70$ | $-22.760 \pm 1.726$ | $-8.419$ | $-0.24 \pm 0.04$ | $-2.99 \pm 0.52$ | $1.48 \pm 0.59$ |
| 43 | Siberian Shelves | Polar | 1848 | 1889 | $-0.47 \pm 0.10$ | $-0.90$ | $-10.499 \pm 2.117$ | $-20.322$ | $-0.05 \pm 0.01$ | $-1.09 \pm 0.28$ | $4.10 \pm 0.64$ |
| 44 | Barents and Kara Seas | Polar | 1559 | 1680 | $-0.75 \pm 0.14$ | $-1.60$ | $-14.176 \pm 2.585$ | $-32.225$ | $-0.11 \pm 0.02$ | $-2.05 \pm 0.43$ | $1.58 \pm 0.46$ |
| 45 | Antarctic Shelves | Polar | 2452 | 2936 | $-0.90 \pm 0.14$ | $-0.15$ | $-26.630 \pm 3.989$ | $-5.381$ | $-0.69 \pm 0.07$ | $-20.30 \pm 2.18$ | $2.08 \pm 0.29$ |

[revised manuscript text omitted]